# NONPARAMETRIC DATA ATTRIBUTION FOR DIFFUSION MODELS

## ABSTRACT

Data attribution for generative models seeks to quantify the influence of individual training examples on model outputs. Existing methods for diffusion models typically require access to model gradients or retraining, limiting their applicability in proprietary or large-scale settings. We propose a *nonparametric* attribution method that operates entirely on data, measuring influence via patch-level similarity between generated and training images. Our approach is grounded in the analytical form of the optimal score function and naturally extends to multiscale representations, while remaining computationally efficient through convolution-based acceleration. In addition to producing spatially interpretable attributions, our framework uncovers patterns that reflect intrinsic relationships between training data and outputs, independent of any specific model. Experiments demonstrate that our method achieves strong attribution performance, closely matching gradient-based approaches and substantially outperforming existing nonparametric baselines.

## 1 INTRODUCTION

Recent advances in generative models, particularly diffusion models (Ho et al., 2020; Song et al., 2021), have led to significant progress in image synthesis and editing (Rombach et al., 2022; Ramesh et al., 2022; Meng et al., 2022). As these powerful models are trained on increasingly large-scale datasets that often contain private, copyrighted, or low-quality content, concerns around data transparency, accountability, and ethical use are growing (Carlini et al., 2023; Saveri & Butterick, 2023). These concerns motivate the study of *data attribution*: identifying the influence of individual training examples on a given generation. Effective data attribution not only supports the responsible deployment of generative models but also enables various downstream applications, including interpreting model behavior (Koh & Liang, 2017; Sui et al., 2021; Ilyas et al., 2022), detecting mislabeled or poisoned data (Jia et al., 2021), guiding data valuation (Nohyun et al., 2023), and improving dataset quality through informed curation (Khanna et al., 2018; Jia et al., 2021; Liu et al., 2021).

Much progress has been made in attributing image generations to training data. Retraining-based methods (Ghorbani & Zou, 2019; Ilyas et al., 2022) assess how generations change when specific training data are removed. While effective, these methods typically require retraining the model tens of thousands of times on different data subsets (Ghorbani & Zou, 2019), making them computationally expensive. To improve efficiency, recent works (Zheng et al., 2024; Lin et al., 2025; Mlodozeniec et al., 2025) adopt approximations based on additive attribution scores (Park et al., 2023a), enabling scalable attribution on large datasets. A common assumption in these methods is access to model gradients, i.e., full access to the generative model. However, this assumption is not always practical. For example, in scenarios where users seek copyright protection against infringement by proprietary models (Somepalli et al., 2023; Zhao et al., 2024), the model gradients may not be accessible. Furthermore, when attribution is intended to support tasks such as data selection (Gu et al., 2025), training a generative model solely for attribution may be prohibitively costly or infeasible.

These challenges call for an attribution method that does not require access to the generative model, either to handle proprietary or black-box models or to serve as a fast surrogate without the cost of training or accessing the model. Existing methods (Zheng et al., 2024; Mlodozeniec et al., 2025) typically measure the similarity between generated images and training data in feature spaces, but often perform poorly as they disregard the behavior of the generative model. Effective attribution for diffusion models in this restricted-access setting remains an open challenge.

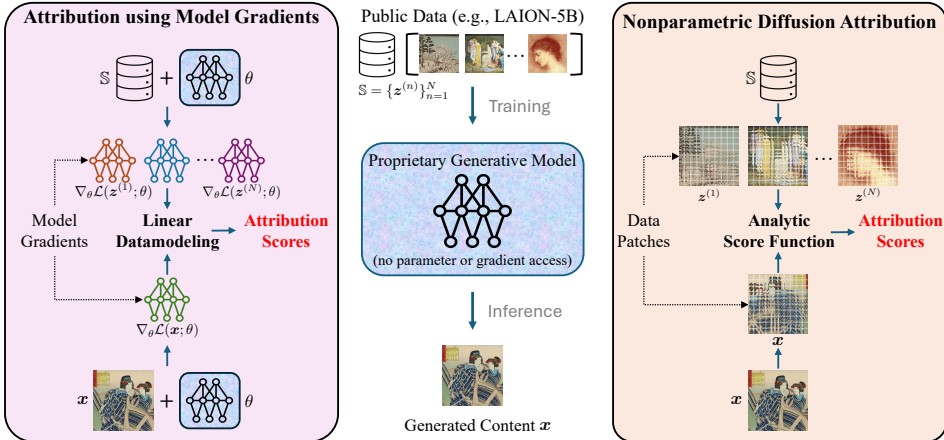

Figure 1: Schematic illustration of attribution methods. **Left:** Model-based attribution relies on gradients and requires parameter access. **Right:** Our *Nonparametric Diffusion Attribution* (**NDA**) compares local image patches via an analytic score function, enabling attribution without model access.

In this paper, we present a *nonparametric* approach to image data attribution in diffusion models, which directly quantifies the influence of training samples on generated outputs through patch-level comparisons. Drawing inspiration from analytical expressions of the score function in diffusion models (Gu et al., 2024; Kamb & Ganguli, 2024), we extract local patches from both generated images and training samples and compute attribution scores based on pairwise distances, as illustrated in Figure 1. Unlike prior methods that require retraining or gradient access, our method operates entirely on data, without any reliance on model architecture or training. Moreover, it provides fine-grained, patch-level attribution, enabling the localization of influential training regions and offering interpretable insights into the generation behaviors. Due to space constraints, related work is deferred to Appendix C.

## 2 PRELIMINARIES

Our approach is grounded in the optimal score function of diffusion models. In this section, we begin with a brief overview of diffusion models and their optimal empirical score functions, followed by a review of data attribution techniques and evaluation protocols.

### 2.1 DIFFUSION MODELS

Diffusion models (Ho et al., 2020; Song et al., 2021) are a class of probabilistic generative models that learn to approximate a data distribution $q(\boldsymbol{x})$ by modeling a parametrized Markovian process $p_\theta(\boldsymbol{x})$. Specifically, it defines a forward process that transforms a clean data sample $\boldsymbol{x} \sim q(\boldsymbol{x})$ into a noisy sequence $\boldsymbol{x}_{1:T} = \boldsymbol{x}_1, \cdots, \boldsymbol{x}_T$ by gradually adding Gaussian noise. The transition probability is given by $q\left(\boldsymbol{x}_t|\boldsymbol{x}_{t-1}\right) \triangleq \mathcal{N}\left(\boldsymbol{x}_t|\sqrt{1-\beta_t}\boldsymbol{x}_{t-1}, \beta_t\mathbf{I}\right)$, where $\{\beta_t\}_{t=1}^T$ denotes a predefined variance schedule. A key property of the forward process is that $\boldsymbol{x}_t$ can be sampled in closed form at any timestep $t$:

$$q(\boldsymbol{x}_t|\boldsymbol{x}) = \mathcal{N}(\boldsymbol{x}_t|\sqrt{\bar{\alpha}_t}\boldsymbol{x}, (1-\bar{\alpha}_t)\mathbf{I}), \tag{1}$$

where $\bar{\alpha}_t \triangleq \prod_{s=1}^t \alpha_s$ and $\alpha_t \triangleq 1 - \beta_t$. As $t$ increases from $0$ to $T$, the noise level increases and $\bar{\alpha}_t$ decays from $1$ to $0$. Consequently, the marginal distribution $q_t(\boldsymbol{x}_t)$ approaches a standard Gaussian, i.e., $q_T(\boldsymbol{x}_T) \approx \mathcal{N}(0, \mathbf{I})$. The reverse process is learned by approximating the reverse conditionals $q(\boldsymbol{x}_{t-1}|\boldsymbol{x}_t)$ using a neural network model $p_\theta(\boldsymbol{x}_{t-1}|\boldsymbol{x}_t) \triangleq \mathcal{N}(\boldsymbol{x}_{t-1}|\boldsymbol{\mu}_\theta(\boldsymbol{x}_t, t), \sigma_t^2\mathbf{I})$, where the variance $\sigma_t^2$ are typically chosen as hyperparameters (Bao et al., 2022). In practice, diffusion models are usually trained to predict the added noise $\boldsymbol{\epsilon}_\theta(\boldsymbol{x}_t, t)$, which relates to the mean via $\boldsymbol{\mu}_\theta(\boldsymbol{x}_t, t) = \frac{1}{\sqrt{\alpha_t}}(\boldsymbol{x}_t - \frac{\beta_t}{\sqrt{1-\bar{\alpha}_t}}\boldsymbol{\epsilon}_\theta(\boldsymbol{x}_t, t))$ (Ho et al., 2020). The model is learned by minimizing a variational bound on the negative log-likelihood:

$$\mathcal{L}_{\text{ELBO}}(\boldsymbol{x}; \theta) = \mathbb{E}_{\boldsymbol{\epsilon}, t}\left[\frac{\beta_t^2}{2\sigma_t^2\alpha_t\left(1-\bar{\alpha}_t\right)}\left\|\boldsymbol{\epsilon} - \boldsymbol{\epsilon}_\theta\left(\sqrt{\bar{\alpha}_t}\boldsymbol{x} + \sqrt{1-\bar{\alpha}_t}\boldsymbol{\epsilon}, t\right)\right\|_2^2\right], \tag{2}$$

where $\epsilon \sim \mathcal{N}(\epsilon|\mathbf{0}, \mathbf{I})$. Let $\mathbb{S} \triangleq \{\boldsymbol{z}^{(n)}|\boldsymbol{z}^{(n)} \sim q\}_{n=1}^{N}$ be the training dataset. The empirical training objective is then $\mathcal{L}_{\text{ELBO}}(\mathbb{S}; \theta) = \frac{1}{N}\sum_{n=1}^{N} \mathcal{L}_{\text{ELBO}}\left(\boldsymbol{z}^{(n)}; \theta\right)$. To enhance sample quality, a simplified training objective is often used (Ho et al., 2020):

$$\mathcal{L}_{\text{Simple}}(\boldsymbol{x}; \theta) = \mathbb{E}_{\epsilon, t}\left[\|\epsilon_\theta\left(\boldsymbol{x}_t, t\right) - \epsilon\|^2\right], \tag{3}$$

with the empirical objective on $\mathbb{S}$ given by $\mathcal{L}_{\text{Simple}}(\mathbb{S}; \theta) = \frac{1}{N}\sum_{n=1}^{N} \mathcal{L}_{\text{Simple}}(\boldsymbol{z}^{(n)}; \theta)$.

## 2.2 OPTIMAL SCORE FUNCTIONS

A key property (Song et al., 2021; Kingma & Gao, 2023) of diffusion models is that the optimal noise prediction function $\epsilon^*(\boldsymbol{x}_t, t)$ is closely related to the *score function* $\boldsymbol{s}(\boldsymbol{x}_t, t) \triangleq \nabla_{\boldsymbol{x}_t} \log q_t(\boldsymbol{x}_t)$ via:

$$\boldsymbol{s}(\boldsymbol{x}_t, t) = -\frac{\epsilon^*(\boldsymbol{x}_t, t)}{\sqrt{1 - \bar{\alpha}_t}}. \tag{4}$$

Intuitively, for a finite dataset $\mathbb{S} = \{\boldsymbol{z}^{(n)}\}_{n=1}^{N}$, the marginal distribution $q_t\left(\boldsymbol{x}_t\right)$ becomes a Gaussian mixture centered at the scaled data points $\sqrt{\bar{\alpha}_t}\boldsymbol{z}^{(n)}$: $q_t(\boldsymbol{x}_t) = \frac{1}{N}\sum_{n=1}^{N} \mathcal{N}\left(\boldsymbol{x}_t|\sqrt{\bar{\alpha}_t}\boldsymbol{z}^{(n)}, (1 - \bar{\alpha}_t)\mathbf{I}\right)$. The score function $\boldsymbol{s}(\boldsymbol{x}_t, t)$ therefore admits an analytical form (see Appendix B for details):

$$\boldsymbol{s}(\boldsymbol{x}_t, t) = \nabla_{\boldsymbol{x}_t} \log q_t(\boldsymbol{x}_t) = \frac{1}{1 - \bar{\alpha}_t}\sum_{n=1}^{N}(\sqrt{\bar{\alpha}_t}\boldsymbol{z}^{(n)} - \boldsymbol{x}_t)W_t(\boldsymbol{z}^{(n)}|\boldsymbol{x}_t), \tag{5}$$

where $W_t(\boldsymbol{z}^{(n)}|\boldsymbol{x}_t)$ is a weighting term defined as:

$$W_t(\boldsymbol{z}^{(n)}|\boldsymbol{x}_t) = \frac{\mathcal{N}\left(\boldsymbol{x}_t|\sqrt{\bar{\alpha}_t}\boldsymbol{z}^{(n)}, (1 - \bar{\alpha}_t)\mathbf{I}\right)}{\sum_{n'=1}^{N} \mathcal{N}\left(\boldsymbol{x}_t|\sqrt{\bar{\alpha}_t}\boldsymbol{z}^{(n')}, (1 - \bar{\alpha}_t)\mathbf{I}\right)}. \tag{6}$$

This score function $\boldsymbol{s}(\boldsymbol{x}_t, t)$ can be interpreted as a conditional average over the added noise, where the residual term $\boldsymbol{x}_t - \sqrt{\bar{\alpha}_t}\boldsymbol{z}^{(n)}$ is averaged across training examples, weighted by the posterior probability $W_t(\boldsymbol{z}^{(n)}|\boldsymbol{x}_t)$ that $\boldsymbol{x}_t$ was transformed from $\boldsymbol{z}^{(n)}$ at $t = 0$ under the forward process.

## 2.3 DATA ATTRIBUTION AND EVALUATION METRICS

Given a training dataset $\mathbb{S}$ and a generated sample $\boldsymbol{x}$, the goal of data attribution is to quantify the influence of each training example in $\mathbb{S}$ on the generation of $\boldsymbol{x}$. Formally, this involves assigning an attribution score $\tau(\boldsymbol{x}, \boldsymbol{z}^{(n)}; \mathbb{S})$ to each training example, reflecting its relative importance in generating $\boldsymbol{x}$.

We follow Zheng et al. (2024) and adopt the linear datamodeling score (LDS; Park et al., 2023a) as our evaluation metric, which quantifies how well an attribution method aligns with the ground-truth influence of training data on model outputs. Specifically, for a training dataset $\mathbb{S} = \{\boldsymbol{z}^{(n)}\}_{n=1}^{N}$ of size $N$, LDS evaluates an attribution method $\tau$ by first sampling multiple random subsets $\{\mathbb{S}_m \subset \mathbb{S}\}_{m=1}^{M}$. Let $\theta^*(\mathbb{S}_m)$ denote the generative model trained on subset $\mathbb{S}_m$, and let $\mathcal{F}(\boldsymbol{x}; \theta^*(\mathbb{S}_m))$ denote the model output on a test input $\boldsymbol{x}$.[1] Then, for each subset $\mathbb{S}_m$, the attribution scores $\tau(\boldsymbol{x}, \boldsymbol{z}^{(n)}; \mathbb{S})$ assigned to training examples in $\mathbb{S}_m$ are summed to form an attribution-based prediction $g_\tau(\boldsymbol{x}, \mathbb{S}_m; \mathbb{S})$:

$$g_\tau(\boldsymbol{x}, \mathbb{S}_m; \mathbb{S}) \triangleq \sum_{\boldsymbol{z}^{(n)} \in \mathbb{S}_m} \tau(\boldsymbol{x}, \boldsymbol{z}^{(n)}; \mathbb{S}). \tag{7}$$

Finally, the LDS score for attribution method $\tau$ on test input $\boldsymbol{x}$ is calculated as the Spearman rank correlation $\rho(\cdot, \cdot)$ between the ground-truth outputs from the $M$ retrained models (each trained on a different subset $\mathbb{S}_m$) and the corresponding attribution-based predictions:

$$\text{LDS}(\tau, \boldsymbol{x}) \triangleq \rho\Big(\{\mathcal{F}(\boldsymbol{x}; \theta^*(\mathbb{S}_m)) : m \in [M]\}, \{g_\tau(\boldsymbol{x}, \mathbb{S}_m; \mathbb{S}) : m \in [M]\}\Big). \tag{8}$$

---

[1]For diffusion models, we set $\mathcal{F} = \mathcal{L}_{\text{Simple}}$, representing the model output used in LDS evaluation.

## 3 METHODOLOGY

Our approach is motivated by the weighting term in the optimal score function of diffusion models, which naturally encodes the relative importance of each training example during generation. Leveraging this insight, we build upon the recent theoretical framework of Kamb & Ganguli (2024), which extends the analysis of optimal score functions to capture inductive biases that promote generalization in diffusion models. By bridging this framework with the problem of data attribution, we develop a nonparametric data attribution method that does not require access to model gradients or retraining.

### 3.1 EQUIVARIANT AND LOCAL SCORE MACHINES

While Eq. (5) defines the optimal score function, it relies solely on distance-based weighting in image space and does not generalize beyond the training set. Kamb & Ganguli (2024) derive an analytical form under inductive biases of *locality* and *equivariance*. This formulation preserves the structure of the empirical score but yields meaningful similarity by incorporating spatial structure and symmetry.

Let $\boldsymbol{x}_t \in \mathbb{R}^{C \times L \times L}$ denote a noisy image at diffusion time $t$, with $C$ channels and spatial resolution $L$. For a pixel location $\ell \in [L] \times [L]$, let $\boldsymbol{x}_{t,\ell} \in \mathbb{R}^C$ denote its pixel value. Define $\Omega_\ell$ as the $P \times P$ neighborhood centered at $\ell$, and $\boldsymbol{x}_{t,\Omega_\ell} \in \mathbb{R}^{C \times P \times P}$ as the corresponding local patch. Let $\mathbb{P}_\Omega(\mathbb{S})$ be the set of all such patches extracted from the training dataset $\mathbb{S}$. Each patch $\boldsymbol{u} \in \mathbb{P}_\Omega(\mathbb{S})$ is thus a local crop of some training image $\boldsymbol{z} \in \mathbb{S}$, and we denote its center pixel as $\boldsymbol{u}_0$. Under the locality and equivariance assumptions, Kamb & Ganguli (2024) show that the optimal MMSE estimator of the score function at pixel location $\ell$, denoted by $\boldsymbol{s}(\boldsymbol{x}_t, t, \ell) \in \mathbb{R}^C$, takes the form:

$$\boldsymbol{s}(\boldsymbol{x}_t, t, \ell) = \sum_{\boldsymbol{u} \in \mathbb{P}_\Omega(\mathbb{S})} \frac{\sqrt{\bar{\alpha}_t}\boldsymbol{u}_0 - \boldsymbol{x}_{t,\ell}}{1 - \bar{\alpha}_t} W_t(\boldsymbol{u}|\boldsymbol{x}_{t,\Omega_\ell}), \tag{9}$$

where the weighting term $W_t(\boldsymbol{u}|\boldsymbol{x}_{t,\Omega_\ell})$ is defined as:

$$W_t(\boldsymbol{u}|\boldsymbol{x}_{t,\Omega_\ell}) = \frac{\mathcal{N}\left(\boldsymbol{x}_{t,\Omega_\ell}|\sqrt{\bar{\alpha}_t}\boldsymbol{u}, (1 - \bar{\alpha}_t)\,\mathbf{I}\right)}{\sum_{\boldsymbol{v} \in \mathbb{P}_\Omega(\mathbb{S})} \mathcal{N}\left(\boldsymbol{x}_{t,\Omega_\ell}|\sqrt{\bar{\alpha}_t}\boldsymbol{v}, (1 - \bar{\alpha}_t)\,\mathbf{I}\right)}. \tag{10}$$

This formulation generalizes Eq. (5) by measuring similarity at the patch level rather than over full images, allowing it to exploit fine-grained local structure.

### 3.2 PATCH-BASED DATA ATTRIBUTION SCORES

The weighting term in Eq. (10) can be expressed as a softmax over quadratic distances $\frac{\|\boldsymbol{x}_{t,\Omega_\ell} - \sqrt{\bar{\alpha}_t}\boldsymbol{u}\|^2}{2(1 - \bar{\alpha}_t)}$, which naturally reflects the contribution of each training patch $\boldsymbol{u}$ to the generation of pixel $\boldsymbol{x}_{t,\ell}$. We reinterpret this term as a *patch-wise influence* score that quantifies the influence of a training patch $\boldsymbol{u}$ on the local region of a generated image. By aggregating these local scores across spatial locations, we obtain a nonparametric, spatially interpretable attribution measure.

**Patch-wise influence.** For a noisy patch $\boldsymbol{x}_{t,\Omega_\ell}$ centered at $\ell$ in a generated image $\boldsymbol{x}$ at timestep $t$, we adopt the local weighting from Eq. (10) and define the patch-wise influence score as:

$$\tau(\boldsymbol{x}_{t,\Omega_\ell}, \boldsymbol{u}; \mathbb{P}_\Omega(\mathbb{S})) \triangleq \exp\left(-\frac{\|\boldsymbol{x}_{t,\Omega_\ell} - \sqrt{\bar{\alpha}_t}\boldsymbol{u}\|^2}{2(1 - \bar{\alpha}_t)}\right) \cdot \left(\sum_{\boldsymbol{v} \in \mathbb{P}_\Omega(\mathbb{S})} \exp\left(-\frac{\|\boldsymbol{x}_{t,\Omega_\ell} - \sqrt{\bar{\alpha}_t}\boldsymbol{v}\|^2}{2(1 - \bar{\alpha}_t)}\right)\right)^{-1}, \tag{11}$$

which is a normalized similarity score over all patches in $\mathbb{P}_\Omega(\mathbb{S})$ extracted from the training set.

**Image-level attribution.** Given a generated image $\boldsymbol{x}$, we apply the forward diffusion process to obtain noisy samples $\boldsymbol{x}_t$. For each pixel location $\ell$, we extract the $P \times P$ patch $\boldsymbol{x}_{t,\Omega_\ell}$ from $\boldsymbol{x}_t$, using zero-padding near boundaries to handle incomplete patches. The patch-wise influence scores in Eq. (11) is then computed for all training patches $\boldsymbol{u} \in \mathbb{P}_\Omega(\mathbb{S})$.

To aggregate into an image-level attribution score $\tau(\boldsymbol{x}, \boldsymbol{z}^{(n)}; \mathbb{S})$, we proceed as follows: (1) For each patch $\boldsymbol{x}_{t,\Omega_\ell}$ of $\boldsymbol{x}_t$, we select the $k$ most influential patches from each training image $\boldsymbol{z}^{(n)}$, denoted by

$\mathbb{P}_\Omega^k(\boldsymbol{x}_{t,\Omega_\ell}, \boldsymbol{z}^{(n)})$.[2] (2) We sum the influence scores of these top-$k$ patches to estimate the contribution of the training image $\boldsymbol{z}^{(n)}$ to generating the local region around $\ell$. (3) Finally, we aggregate across all spatial locations $\ell$ and average over a set of timesteps $\mathcal{T}$:

$$\tau(\boldsymbol{x}, \boldsymbol{z}^{(n)}; \mathbb{S}) \triangleq \frac{1}{|\mathcal{T}|} \sum_{t \in \mathcal{T}} \sum_{\ell} \sum_{\boldsymbol{u} \in \mathbb{P}_\Omega^k(\boldsymbol{x}_{t,\Omega_\ell}, \boldsymbol{z}^{(n)})} \tau(\boldsymbol{x}_{t,\Omega_\ell}, \boldsymbol{u}; \mathbb{P}_\Omega(\mathbb{S})). \tag{12}$$

This attribution score is both *spatially interpretable*, as it aggregates patch-wise influence with local meaning, and *nonparametric*, as it operates entirely on training data without relying on the model.

### 3.3 ATTRIBUTION WITH MULTISCALE PATCH-WISE INFLUENCE

The attribution method described above relies on quadratic Euclidean distances between fixed-size patches in the original image space. However, images may have varying resolutions, and a single patch size may fail to capture different levels of information: from fine-grained textures to higher-level structures (Adelson et al., 1984; Lin et al., 2017). Moreover, diffusion models are known to generate different levels of detail across timesteps: early (high-noise) stages capture coarse structures, while later (low-noise) stages refine local details (Jing et al., 2022; Park et al., 2023b).

To account for these effects, we introduce a *multiscale* extension that computes patch-wise influence across multiple resolutions. Specifically, we downsample both generated and training patches and evaluate distances in the lower-resolution space. Let $\mathrm{D}(\cdot)$ denote a downsampling operator, and define $\widehat{\boldsymbol{x}}_{t,\Omega_\ell} \triangleq \mathrm{D}(\boldsymbol{x}_{t,\Omega_\ell})$, $\widehat{\boldsymbol{u}} \triangleq \mathrm{D}(\boldsymbol{u})$, and $\widehat{\boldsymbol{v}} \triangleq \mathrm{D}(\boldsymbol{v})$. The low-resolution patch-wise influence score is then:

$$\widehat{\tau}(\boldsymbol{x}_{t,\Omega_\ell}, \boldsymbol{u}; \mathbb{P}_\Omega(\mathbb{S})) \triangleq \exp\left( -\frac{\|\widehat{\boldsymbol{x}}_{t,\Omega_\ell} - \sqrt{\bar{\alpha}_t}\widehat{\boldsymbol{u}}\|^2}{2(1 - \bar{\alpha}_t)} \right) \cdot \left( \sum_{\boldsymbol{v} \in \mathbb{P}_\Omega(\mathbb{S})} \exp\left( -\frac{\|\widehat{\boldsymbol{x}}_{t,\Omega_\ell} - \sqrt{\bar{\alpha}_t}\widehat{\boldsymbol{v}}\|^2}{2(1 - \bar{\alpha}_t)} \right) \right)^{-1}. \tag{13}$$

We then combine the original and low-resolution influence measures into a multiscale score:

$$\tau^{\mathrm{ms}}(\boldsymbol{x}_{t,\Omega_\ell}, \boldsymbol{u}; \mathbb{P}_\Omega(\mathbb{S})) \triangleq \gamma_t \tau(\boldsymbol{x}_{t,\Omega_\ell}, \boldsymbol{u}; \mathbb{P}_\Omega(\mathbb{S})) + (1 - \gamma_t)\widehat{\tau}(\boldsymbol{x}_{t,\Omega_\ell}, \boldsymbol{u}; \mathbb{P}_\Omega(\mathbb{S})), \tag{14}$$

where $\gamma_t \in [0, 1]$ is a timestep-dependent weighting factor that balances the contribution of the original and low-resolution influence. Finally, we extend this to multiscale image-level attribution by aggregating over timesteps $\mathcal{T}$ and spatial locations $\ell$:

$$\tau^{\mathrm{ms}}(\boldsymbol{x}, \boldsymbol{z}^{(n)}; \mathbb{S}) \triangleq \frac{1}{|\mathcal{T}|} \sum_{t \in \mathcal{T}} \sum_{\ell} \sum_{\boldsymbol{u} \in \mathbb{P}_\Omega^k(\boldsymbol{x}_{t,\Omega_\ell}, \boldsymbol{z}^{(n)})} \tau^{\mathrm{ms}}(\boldsymbol{x}_{t,\Omega_\ell}, \boldsymbol{u}; \mathbb{P}_\Omega(\mathbb{S})). \tag{15}$$

### 3.4 CONVOLUTION-BASED ACCELERATION

Directly evaluating Eq. (11 & 13) on large-scale datasets is computationally challenging. In a naive implementation, each test patch $\boldsymbol{x}_{t,\Omega_\ell}$ is broadcast against all unfolded training patches $\boldsymbol{u} \in \mathbb{P}_\Omega(\mathbb{S})$, leading to peak memory consumption of $\mathcal{O}(NL^2CP^2)$, which is $P^2$ times larger than the dataset itself.

To address this, we propose a memory-efficient implementation that avoids explicit patch unfolding by leveraging convolutional operators. Specifically, to compute $\|\boldsymbol{x}_{t,\Omega_\ell} - \sqrt{\bar{\alpha}_t}\boldsymbol{u}\|^2$ for all $L \times L$ patches $\boldsymbol{u}$ from a training image $\boldsymbol{z}^{(n)}$, we treat the patch $\boldsymbol{x}_{t,\Omega_\ell} \in \mathbb{R}^{C \times P \times P}$ as a convolutional kernel. Applying this kernel to the training image yields the inner-products $\langle \boldsymbol{x}_{t,\Omega_\ell}, \boldsymbol{u} \rangle$ over all spatial locations in a single convolution pass. The quadratic distance is then obtained as $\|\boldsymbol{x}_{t,\Omega_\ell}\|^2 - 2\sqrt{\bar{\alpha}_t}\langle \boldsymbol{x}_{t,\Omega_\ell}, \boldsymbol{u} \rangle + \bar{\alpha}_t\|\boldsymbol{u}\|^2$.

This approach leverages GPU-optimized convolutions to reduce memory usage. We further parallelize across the $L^2$ test patches by batching $B$ patches into a convolutional kernel with $B$ output channels. This yields peak memory $\mathcal{O}(BNL^2)$, which does not explicitly scale with patch size $P$, thereby avoiding the prohibitive $P^2$ factor of naive unfolding and enabling scalable attribution on large datasets.

---

[2]Formally, $\mathbb{P}_\Omega^k(\boldsymbol{x}_{t,\Omega_\ell}, \boldsymbol{z}^{(n)}) \subset \mathbb{P}_\Omega(\{\boldsymbol{z}^{(n)}\})$ with $|\mathbb{P}_\Omega^k(\boldsymbol{x}_{t,\Omega_\ell}, \boldsymbol{z}^{(n)})| = k$, and for all $\boldsymbol{u}' \in \mathbb{P}_\Omega^k(\boldsymbol{x}_{t,\Omega_\ell}, \boldsymbol{z}^{(n)})$ and $\boldsymbol{u}'' \in \mathbb{P}_\Omega(\{\boldsymbol{z}^{(n)}\}) \setminus \mathbb{P}_\Omega^k(\boldsymbol{x}_{t,\Omega_\ell}, \boldsymbol{z}^{(n)})$, we have $\tau(\boldsymbol{x}_{t,\Omega_\ell}, \boldsymbol{u}'; \mathbb{P}_\Omega(\mathbb{S})) \geq \tau(\boldsymbol{x}_{t,\Omega_\ell}, \boldsymbol{u}''; \mathbb{P}_\Omega(\mathbb{S}))$.

## 3.5 DISCUSSIONS

Data attribution is often regarded as a lens for understanding model behavior, quantifying the causal influence of individual training examples on model predictions through the learning process. From this perspective, it might seem paradoxical to speak of attribution without specifying a model: if no model is given, what exactly is being attributed?

Our work invites a broader interpretation. In a narrow sense, our nonparametric method can be understood as a principled guess of how training data might have influenced a generated sample, assuming it was produced by some model trained on the same data. Since our method does not access model parameters, it produces the same attribution for all possible models trained on the dataset. At first glance this may appear counterintuitive, because different models can generalize in different ways and might be expected to yield different attribution patterns. However, our empirical results show that the attribution scores remain consistent across a variety of architectures and training regimes. This suggests that there exists a shared, model-agnostic structure to generalization, which reflects intrinsic relationships between training data and outputs, independent of any specific model.

Seen from an even broader perspective, this idea resonates with how humans attribute provenance. People are often able to determine which training images most likely inspired a generated image even without any knowledge of the mechanism that produced it. This form of attribution is grounded in *perceived similarity* rather than *parametric causality*, and it applies whether the image was produced by a neural network, a human artist, or some natural process. In this light, nonparametric data attribution can be viewed as an attempt to formalize this intuitive, model-independent notion of influence: some training examples are simply more responsible for a given generation than others.

## 4 EXPERIMENTS

We evaluate our method, *Nonparametric Diffusion Attribution* (**NDA**), against both nonparametric and gradient-based attribution approaches using two complementary protocols: the linear datamodeling score (LDS), which quantifies alignment with ground-truth influence, and counterfactual evaluation, which assesses the effect of removing influential training data on generated outputs. Our results and ablations show that NDA achieves strong attribution performance without accessing model parameters or gradients, while qualitative visualizations highlight its spatial interpretability and visual consistency.[3]

### 4.1 EXPERIMENTAL SETUP

**Datasets.** We conduct experiments on CIFAR-10 (Krizhevsky, 2009) and CelebA (Liu et al., 2015). For efficient ablation studies, we follow Zheng et al. (2024) and construct a CIFAR-2 subset consisting of 5,000 images from CIFAR-10. Additional dataset details are provided in Appendix D.1.

**Target models.** For each dataset, we train a diffusion model to serve as the target model for data attribution. On CIFAR-2 and CIFAR-10, we adopt the original DDPM implementation (Ho et al., 2020) to train an unconditional diffusion model with a U-Net backbone containing 35.7M parameters. For CelebA, we use the same implementation but with a modified architecture of 118.8M parameters to accommodate the $64 \times 64$ resolution. The number of diffusion steps is fixed to $T{=}1{,}000$. Further training details are provided in Appendix D.2. Note that gradient-based attribution methods directly use the target model parameters, whereas our NDA does not access model parameters or architectures.

**LDS evaluation.** Following Zheng et al. (2024), we sample $M{=}64$ random subsets of the training set $\mathbb{S}$, each containing $50\%$ of the samples. For each subset $\mathbb{S}_m$, we train three models with different random seeds and average their $\mathcal{L}_{\text{Simple}}$ losses as the model output $\mathcal{F}(\boldsymbol{x}; \theta^*(\mathbb{S}_m))$ for a test input $\boldsymbol{x}$. Additional implementation details are provided in Appendix D.3. To ensure a fair comparison, we use the same 1,000-image held-out validation set and 1,000-image generation set as Zheng et al. (2024).

**NDA setup.** For patch-wise influence, we use patch sizes $P \in [3, 21]$ for different timesteps $t \in \mathcal{T}$ and resolutions, as determined by ablation studies in Sec. 4.5. To compute low-resolution patch-wise influence, we apply an average-pooling downsampling operator $\text{D}(\cdot)$ with a window size of 2, reducing patch resolution by half. In all experiments, we select the top-$k$ most influential patches per training image with $k{=}100$ to obtain image-level attribution scores. Due to low signal-to-noise ratios at large timesteps, we restrict the set of timesteps to $\mathcal{T}{=}\{100, 200, 300, 400, 500\}$ in our experiments.

---

[3]Code is available in Supplementary Material.

Table 1: LDS (%) of different attribution methods on CIFAR-2, CIFAR-10, and CelebA.

| Method | CIFAR-2 | | CIFAR-10 | | Celeb-A | |
|---|---|---|---|---|---|---|
| | Validation | Generation | Validation | Generation | Validation | Generation |
| *Without Model Access* | | | | | | |
| Raw pixel (dot prod.) | 7.77±0.57 | 4.89±0.58 | 2.50±0.42 | 2.25±0.39 | 5.58±0.73 | 4.94±1.58 |
| Raw pixel (cosine) | 7.87±0.57 | 5.44±0.57 | 2.71±0.41 | 2.61±0.38 | 6.16±0.75 | 4.38±1.63 |
| CLIP similarity (dot prod.) | 6.51±1.06 | 3.00±0.95 | 2.39±0.41 | 1.11±0.47 | 8.87±1.14 | 2.51±1.13 |
| CLIP similarity (cosine) | 8.54±1.01 | 4.01±0.85 | 3.39±0.38 | 1.69±0.49 | 10.92±0.87 | 3.03±1.13 |
| **NDA (Ours)** | **24.88±0.42** | **15.91±0.49** | **11.81±0.30** | **7.41±0.45** | **16.89±0.59** | **13.92±0.68** |
| *Using Model Gradients* | | | | | | |
| Gradient (dot prod.) | 5.14±0.60 | 2.80±0.55 | 0.79±0.43 | 0.74±0.45 | 3.82±0.50 | 3.83±1.06 |
| Gradient (cosine) | 5.08±0.59 | 2.78±0.54 | 0.66±0.43 | 0.58±0.42 | 3.65±0.52 | 3.86±0.96 |
| TracInCP | 6.26±0.84 | 3.76±0.61 | 0.98±0.44 | 0.96±0.40 | 5.14±0.75 | 5.18±1.05 |
| GAS | 5.78±0.82 | 3.34±0.56 | 0.89±0.48 | 0.90±0.41 | 5.44±0.68 | 4.69±0.97 |
| TRAK | 11.42±0.49 | 5.78±0.48 | 2.93±0.46 | 2.20±0.38 | 11.28±0.47 | 7.02±0.89 |
| D-TRAK | 26.79±0.33 | 18.82±0.43 | 14.69±0.46 | 11.05±0.43 | 22.83±0.51 | 16.84±0.54 |

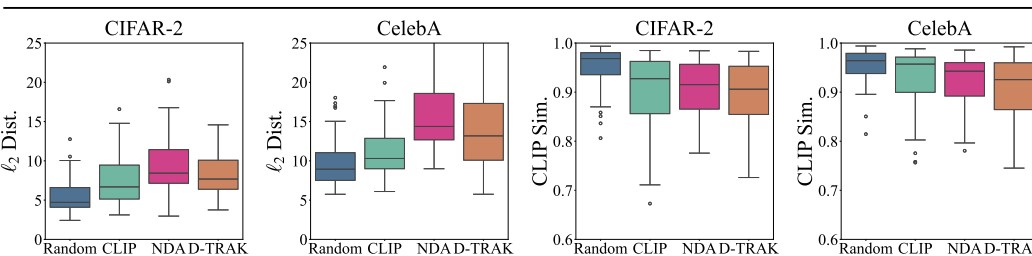

Figure 2: Counterfactual evaluation of $\ell_2$ distance (**Left**) and CLIP similarity (**Right**) between original and regenerated images on CIFAR-2 and CelebA after removing the most influential training samples identified by different attribution methods and retraining the model.

## 4.2 MAIN RESULTS: EVALUATING LDS FOR ATTRIBUTION METHODS

We compare NDA against a range of attribution baselines; detailed descriptions are provided in Appendix D.4. Our primary comparison focuses on representative approaches that do not require access to model parameters, with gradient-based methods included as references. Table 1 reports the results on CIFAR-2, CIFAR-10, and CelebA.

Compared to the strongest baseline based on CLIP cosine similarity, NDA achieves consistent and substantial gains across both validation and generation sets. On CIFAR-2, NDA improves over CLIP by +16.32 (validation) and +11.90 (generation); on CIFAR-10, by +8.42 and +5.72; and on CelebA, by +5.97 and +10.89, respectively. When compared to strong gradient-based methods tailored for diffusion models, such as D-TRAK, NDA substantially closes the gap while accessing no model information. For example, on CelebA, the gap is reduced to 5.94/2.92 (validation/generation), much smaller than the CLIP gap of 11.91/13.81. Overall, NDA approaches the performance of strong parametric baselines while markedly outperforming CLIP similarity on LDS.

## 4.3 COUNTERFACTUAL EVALUATION

To evaluate the faithfulness and practical effectiveness of NDA, we conduct a counterfactual influence experiment on CIFAR-2 and CelebA. For each generated test image, we first identify the top-1,000 most positively influential training samples according to each attribution method, remove these samples from the training set, and retrain the diffusion model from scratch. The test image is then regenerated using the retrained models under the same random seed, and the impact of removal is quantified using pixel-wise $\ell_2$ distance and CLIP cosine similarity. We repeat this procedure for 60 randomly generated test images and report the average results.

We compare NDA against three baselines: Random removal, CLIP similarity (cosine), and D-TRAK. As shown in Figure 2, NDA achieves median $\ell_2$ distances of 8.43 and 14.38 on CIFAR-2 and CelebA, respectively, outperforming CLIP (6.68, 10.30) and D-TRAK (7.68, 13.18). For CLIP similarity, NDA attains 0.92 and 0.94, approaching D-TRAK (0.91, 0.93). These results indicate that NDA effectively identifies and removes training samples with strong influence over the generated output, providing a compelling nonparametric alternative to gradient-based attribution. Qualitative visualizations in Figure 3 further show that the generated images exhibit significant distortions after removing the training samples deemed most influential by NDA.

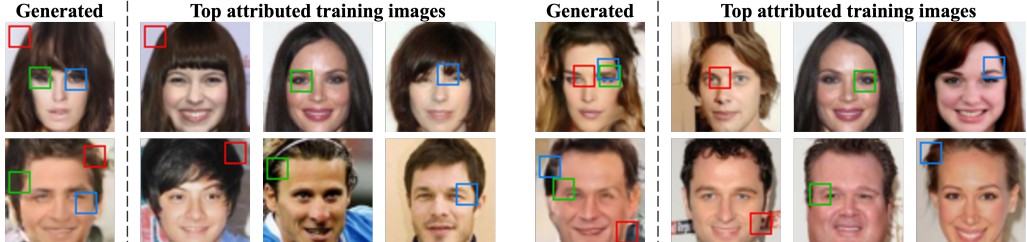

Figure 3: Counterfactual visualization on CIFAR-2 (**Top**) and CelebA (**Bottom**). Images are compared to those generated by retrained models using the same seed. See Appendix F.1 for more cases.

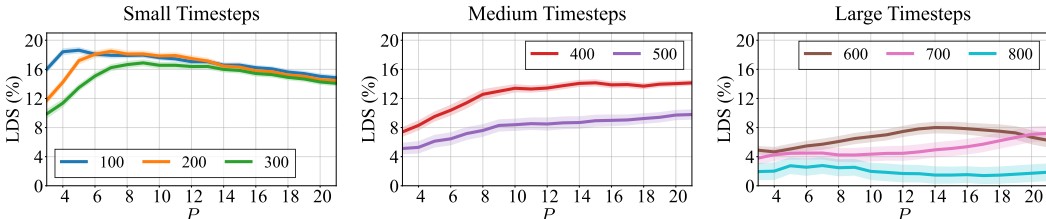

Figure 4: Spatial interpretability of NDA. The patches with the highest patch-wise influence scores (w.r.t. patches in the generated image) are highlighted in the top attributed training images.

Figure 5: LDS (%) on the CIFAR-2 validation set across different patch sizes and timesteps at the original resolution. **Left**: small timesteps ($t \leq 300$), where moderate patch sizes achieve the highest scores. **Middle**: medium timesteps ($t = 400, 500$), where larger patches outperform smaller ones due to higher noise. **Right**: large timesteps, where noise dominates and informative signal is minimal.

## 4.4 SPATIAL INTERPRETABILITY

Since NDA computes attribution by aggregating patch-wise influence, it naturally offers an additional spatial level of interpretability. Intuitively, a training image with a high attribution score must contain patches that strongly align with influential regions of the test image. To visualize this correspondence, we highlight the most influential patches from each of the top attributed training images in Figure 4. We observe that top-ranked training images consistently contain local patches that are visually similar to those in the test image, offering a natural explanation for their high attribution scores.

## 4.5 ABLATION STUDIES

We conduct ablation studies on key hyperparameters of our method. All studies are performed on the validation sets, which are i.i.d. samples drawn from the same distribution as the training sets. The selected hyperparameters are then applied to the generation sets without further tuning to produce the results reported in Sec. 4.2. In all figures where applicable, solid lines show the mean, and shaded regions indicate $\pm 1$ standard deviation.

**Patch size selection.** We study the effect of patch size by evaluating LDS performance across $P \in [3, 21]$ at different timesteps. To isolate the impact of timestep, we fix $\mathcal{T}$ to contain a single timestep (e.g., $\mathcal{T} = \{100\}, \ldots, \{900\}$) and vary $P$. Figure 5 shows LDS curves as a function of $P$ for each $\mathcal{T}$ on CIFAR-2 (see Appendix E for CelebA). We observe that the optimal patch size varies with timestep and generally increases as $t$ grows. At early timesteps ($t \leq 300$), small to moderate patches ($P = 5, 7, 9$) yield the highest LDS scores, suggesting that local patterns dominate when noise levels are relatively low during reverse diffusion. For mid-range timesteps ($400 \leq t \leq 500$), larger patches perform better, due to their ability to aggregate more contextual information under higher noise levels. In the high-noise regime ($t \geq 600$), LDS scores drop significantly across all patch sizes. Based on these results, we adopt a timestep-dependent patch size selection strategy: using smaller patches for early timesteps and progressively larger patches for later timesteps, choosing the $P$ that maximizes LDS for each $t$.

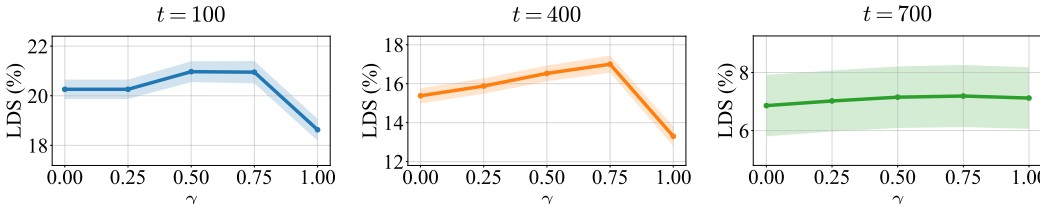

Figure 6: LDS (%) on CIFAR-2 across different weighting factor $\gamma$ at varying timesteps.

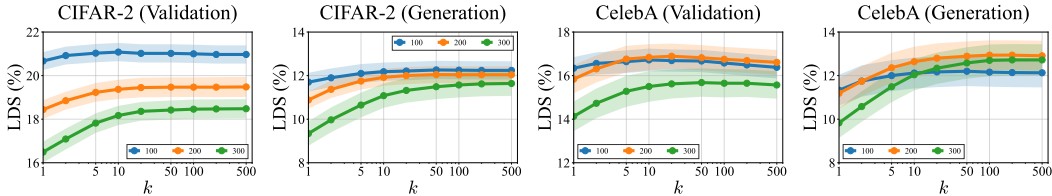

Figure 7: LDS (%) as a function of the number $k$ of top influential patches selected for aggregation on CIFAR-2 (**Left**) and CelebA (**Right**). Setting $k=100$ generally provides strong performance.

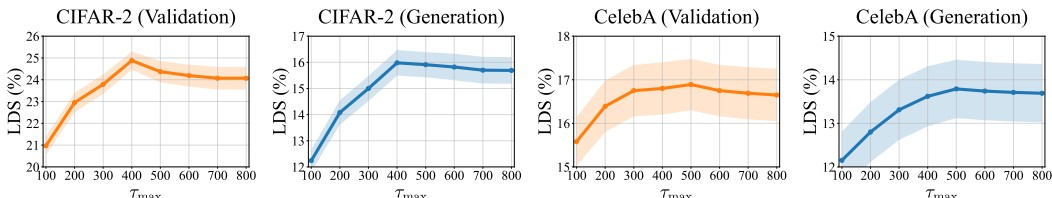

Figure 8: LDS (%) as a function of the max aggregation timestep $\tau_{\max}$ on CIFAR-2 (**Left**) and CelebA (**Right**). Choosing $\tau_{\max}=500$ generally yields good results.

**Multiscale influence.** We first find the optimal patch size for the low-resolution patch-wise influence in Eq. (13) using the same procedure as above and observe a similar trend, where early timesteps favor smaller patches. Interestingly, the optimal patch size differs between the original and low-resolution cases (see Appendix E for details). We then evaluate the proposed multiscale attribution in Eq. (15) by varying the weighting factor $\gamma \in \{0, 0.25, 0.5, 0.75, 1\}$ across timesteps. As shown in Figure 6, combining multiple scales (e.g., $\gamma=0.75$) consistently improves LDS, particularly in the low-noise regime, suggesting that multiscale features provide complementary information that enhances attribution. Further improvements may be possible by incorporating more scales, which we leave for future work.

**Top-$k$ patch selection.** We examine how the number of top-$k$ patches affects attribution. Figure 7 shows LDS as a function of $k$ on lower-noise timesteps, which contribute more significantly to the final attribution. We test $k$ ranging from 1 to 500 and find that an intermediate value of $k=100$ generally achieves optimal performance across timesteps and datasets and generalizes well to generation sets.

**Timestep selection.** Since lower-noise timesteps provide stronger attribution signal, we define $\mathcal{T} = \{100, 200, \ldots, \tau_{\max}\}$ and study the effect of varying the max aggregation timestep $\tau_{\max}$. At each $t \in \mathcal{T}$ we use the optimal patch sizes and weighting factors identified above. Figure 8 shows that enlarging $\mathcal{T}$ improves LDS up to mid-range timesteps, after which gains saturate or degrade. On CIFAR-2, performance rises from $t=100$ to a peak near $\tau_{\max}=400$ before dropping by $\sim 0.5$ at $\tau_{\max} \geq 600$; On CelebA, performance saturates near $\tau_{\max}=400 \sim 500$ with negligible gains beyond. These results suggest that late (high-noise) timesteps contribute little useful signal and may instead inject noise. We therefore fix $\tau_{\max}=500$ for all subsequent experiments, which achieves near-optimal performance while avoiding unnecessary computation and variability.

## 5 CONCLUSION

We present a nonparametric method for data attribution in diffusion models that measures the influence of training examples via patch-level similarity. Our approach leverages the analytical form of the optimal score function, extends to multiscale representations, and remains efficient through convolution-based operations. It provides spatially interpretable attributions and uncovers intrinsic relationships between training data and generated outputs, independent of any specific model. Experiments on CIFAR-2, CIFAR-10, and CelebA show that our method closely matches gradient-based approaches and outperforms existing nonparametric baselines, demonstrating that model-agnostic, data-driven attribution is a practical tool for understanding generative models.

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

## A    LLM USAGE

We used an OpenAI LLM (GPT-5) as a writing and formatting assistant. In particular, it helped refine grammar and phrasing, improve clarity, and suggest edits to figure/table captions and layout (e.g., column alignment, caption length, placement). The LLM did not contribute to research ideation, experimental design, implementation, data analysis, or technical content beyond surface-level edits. All outputs were reviewed and edited by the authors, who take full responsibility for the final text and visuals.

## B    PROOF OF SECTION 3.2

The empirical DDPM objective is

$$
\begin{aligned}
\mathcal{L}_{\text{Simple}}(\mathbb{S};\theta) &= \frac{1}{N}\sum_{n=1}^{N}\mathbb{E}_{t\sim[0,T]}\mathbb{E}_{\boldsymbol{\epsilon}\sim\mathcal{N}(0,\mathbf{I})}\left[\|\boldsymbol{\epsilon}_\theta(\boldsymbol{x}_t,t)-\boldsymbol{\epsilon}\|^2\right] \\
&= \frac{1}{N}\sum_{n=1}^{N}\mathbb{E}_{t\sim[0,T]}\mathbb{E}_{\boldsymbol{\epsilon}\sim\mathcal{N}(0,\mathbf{I})}\left[\|\sqrt{1-\bar{\alpha}_t}s_\theta(\boldsymbol{x}_t,t)+\boldsymbol{\epsilon}\|^2\right] \\
&= \mathbb{E}_{t\sim[0,T]}\mathbb{E}_{\boldsymbol{\epsilon}\sim\mathcal{N}(0,\mathbf{I})}\left[\frac{1}{N}\sum_{n=1}^{N}\|\sqrt{1-\bar{\alpha}_t}s_\theta(\boldsymbol{x}_t,t)+\boldsymbol{\epsilon}\|^2\right] \\
&= \mathbb{E}_{t\sim[0,T]}\left[\int\frac{1}{N}\sum_{n=1}^{N}\|\sqrt{1-\bar{\alpha}_t}s_\theta(\boldsymbol{x}_t,t)+\boldsymbol{\epsilon}\|^2\mathcal{N}(\boldsymbol{\epsilon};0,\mathbf{I})d\boldsymbol{\epsilon}\right].
\end{aligned}
\tag{16}
$$

Since $\boldsymbol{x}_t = \sqrt{\bar{\alpha}_t}\boldsymbol{z}^{(n)}+\sqrt{1-\bar{\alpha}_t}\boldsymbol{\epsilon}$, we have $\boldsymbol{\epsilon} = \frac{\boldsymbol{x}_t-\sqrt{\bar{\alpha}_t}\boldsymbol{z}^{(n)}}{\sqrt{1-\bar{\alpha}_t}}$. Thus $d\boldsymbol{\epsilon}=\frac{d\boldsymbol{x}_t}{\sqrt{1-\bar{\alpha}_t}}$. Then

$$
\mathcal{L}_{\text{Simple}}(\mathbb{S};\theta)=\mathbb{E}_{t\sim[0,T]}\left[\int\frac{1}{N}\sum_{n=1}^{N}\|\sqrt{1-\bar{\alpha}_t}s_\theta(\boldsymbol{x}_t,t)+\frac{\boldsymbol{x}_t-\sqrt{\bar{\alpha}_t}\boldsymbol{z}^{(n)}}{\sqrt{1-\bar{\alpha}_t}}\|^2\mathcal{N}(\boldsymbol{x}_t;\sqrt{\bar{\alpha}_t}\boldsymbol{z}^{(n)},(1-\bar{\alpha}_t)\mathbf{I})\sqrt{1-\bar{\alpha}_t}d\boldsymbol{x}\right].
\tag{17}
$$

The minimization of the objective $\mathcal{L}_{\text{Simple}}(\mathbb{S};\theta)$ can be solved by taking the gradient w.r.t. $s_\theta(\boldsymbol{x}_t,t)$:

$$
\begin{aligned}
\mathbf{0} &= \nabla_{s_\theta(\boldsymbol{x}_t,t)}\left[\frac{1}{N}\sum_{n=1}^{N}\|\sqrt{1-\bar{\alpha}_t}s_\theta(\boldsymbol{x}_t,t)+\frac{\boldsymbol{x}_t-\sqrt{\bar{\alpha}_t}\boldsymbol{z}^{(n)}}{\sqrt{1-\bar{\alpha}_t}}\|^2\mathcal{N}(\boldsymbol{x}_t;\sqrt{\bar{\alpha}_t}\boldsymbol{z}^{(n)},(1-\bar{\alpha}_t)\mathbf{I})\right] \\
&= \sum_{n=1}^{N}2\left[s_\theta(\boldsymbol{x}_t,t)+\frac{\boldsymbol{x}_t-\sqrt{\bar{\alpha}_t}\boldsymbol{z}^{(n)}}{1-\bar{\alpha}_t}\right]\mathcal{N}(\boldsymbol{x}_t;\sqrt{\bar{\alpha}_t}\boldsymbol{z}^{(n)},(1-\bar{\alpha}_t)\mathbf{I}) \\
&= \left[\sum_{n=1}^{N}\mathcal{N}(\boldsymbol{x}_t;\sqrt{\bar{\alpha}_t}\boldsymbol{z}^{(n)},(1-\bar{\alpha}_t)\mathbf{I})\right]s_\theta(\boldsymbol{x}_t;t)+\sum_{n=1}^{N}\mathcal{N}(\boldsymbol{x}_t;\sqrt{\bar{\alpha}_t}\boldsymbol{z}^{(n)},(1-\bar{\alpha}_t)\mathbf{I})\frac{\boldsymbol{x}_t-\sqrt{\bar{\alpha}_t}\boldsymbol{z}^{(n)}}{1-\bar{\alpha}_t}.
\end{aligned}
$$

Then the optimal diffusion model can be written as:

$$
\begin{aligned}
s^*(\boldsymbol{x}_t;t) &= \frac{\sum_{n=1}^{N}\mathcal{N}(\boldsymbol{x}_t;\sqrt{\bar{\alpha}_t}\boldsymbol{z}^{(n)},(1-\bar{\alpha}_t)\mathbf{I})\frac{\sqrt{\bar{\alpha}_t}\boldsymbol{z}^{(n)}-\boldsymbol{x}_t}{1-\bar{\alpha}_t}}{\sum_{n=1}^{N}\mathcal{N}(\boldsymbol{x}_t;\sqrt{\bar{\alpha}_t}\boldsymbol{z}^{(n)},(1-\sqrt{\bar{\alpha}_t})\mathbf{I})} = \frac{\sum_{n=1}^{N}\exp\left(-\frac{\|\boldsymbol{x}_t-\sqrt{\bar{\alpha}_t}\boldsymbol{z}^{(n)}\|^2}{2(1-\bar{\alpha}_t)}\right)\frac{\sqrt{\bar{\alpha}_t}\boldsymbol{z}^{(n)}-\boldsymbol{x}_t}{1-\bar{\alpha}_t}}{\sum_{n=1}^{N}\exp\left(-\frac{\|\boldsymbol{x}_t-\sqrt{\bar{\alpha}_t}\boldsymbol{z}^{(n)}\|^2}{2(1-\bar{\alpha}_t)}\right)} \\
&= \sum_{n=1}^{N}\mathbb{S}\left(-\frac{\|\boldsymbol{x}_t-\sqrt{\bar{\alpha}_t}\boldsymbol{z}^{(n)}\|^2}{2(1-\bar{\alpha}_t)}\right)\frac{\sqrt{\bar{\alpha}_t}\boldsymbol{z}^{(n)}-\boldsymbol{x}_t}{1-\bar{\alpha}_t} = \sum_{n=1}^{N}\frac{\sqrt{\bar{\alpha}_t}\boldsymbol{z}^{(n)}-\boldsymbol{x}_t}{1-\bar{\alpha}_t}W_t(\boldsymbol{z}^{(n)}|\boldsymbol{x}_t).
\end{aligned}
$$

## C    RELATED WORKS

### C.1    DATA ATTRIBUTION

Training data plays a critical role in shaping the behavior of machine learning models. Data attribution aims to measure the contribution of individual training samples to model predictions. Existing methods can be broadly categorized into two classes: retraining-based and retraining-free approaches.

Retraining-based methods, such as empirical influence functions (Feldman & Zhang, 2020), Shapley value estimators (Jia et al., 2021), and datamodels (Ilyas et al., 2022), provide high-fidelity attributions by measuring the effect of removing or modifying each training example. However, these methods typically require retraining the model tens of thousands of times on different data subsets to achieve reliable results, making them computationally infeasible for large-scale settings. Retraining-free methods aim to approximate influence scores without additional retraining, offering a more scalable alternative. These approaches fall into two main categories. The first comprises gradient-based methods without kernelization, which rely solely on first-order gradient signals and avoid computing second-order derivatives. Representative approaches include Gradient (Charpiat et al., 2019), TracInCP (Pruthi et al., 2020) and GAS (Hammoudeh & Lowd, 2022). The second category includes gradient-based methods with kernelization, which incorporate curvature information by constructing kernels, typically using the inverse of the Hessian matrix. However, Hessian inversion is often numerically unstable. To mitigate this, recent work approximates the Hessian with the Fisher information matrix (FIM). For instance, TRAK (Park et al., 2023a) introduces a kernel-based approximation that efficiently estimates influence scores via random projections, which is both accurate and computationally tractable for large-scale models. D-TRAK (Zheng et al., 2024) further improves attribution performance by modifying the output function and training loss in TRAK, resulting in more effective LDS scores. Based on empirical analysis, D-TRAK recommends a specific configuration that achieves the best overall performance.

## C.2 Nonparametric data attribution methods

While most data attribution methods rely on parametric models and require gradient computations or model retraining, an alternative line of research explores nonparametric approaches that operate directly on training data without fitting parametric functions. These are typically similarity-based approaches, which estimate the influence of a training sample based on its similarity to the target using a predefined metric. For example, one can leverage pretrained vision-language embeddings such as CLIP (Radford et al., 2021) features to measure high-level semantic similarity of images (Zheng et al., 2024; Mlodozeniec et al., 2025), or compare raw pixel values directly. However, these methods overlook the internal dynamics of the generative model. As a result, effective data attribution for diffusion models in settings without model access remains an open and challenging problem.

# D Implementation details

## D.1 Datasets

**CIFAR-10** ($32 \times 32$). The CIFAR-10 dataset contains $5,0000$ training samples. For LDS evaluation, we randomly sample $1,000$ validation images from the CIFAR-10 test set. To reduce computational overhead, we also construct a subset called CIFAR-2, consisting of $5,000$ training images randomly selected from the "automobile" and "horse" in CIFAR-10's training set, along with $1,000$ validation images from the corresponding classes in the test set.

**CelebA** ($64 \times 64$). We constructed our dataset by selecting $5,000$ training samples and $1,000$ validation samples from the original training and test splits of CelebA (Liu et al., 2015). Following the preprocessing steps outlined by (Song et al., 2021), we first center crop the images to 140×140 pixels, and then resize them to $64 \times 64$ pixels.

## D.2 Models

**CIFAR.** We follow the original implementation of the unconditional DDPM (Ho et al., 2020), which uses a U-Net backbone comprising approximately 35.7M parameters. The model is trained for 200 epochs with a batch size of $128$, using a cosine annealing learning rate schedule. Additional model configurations include a dropout rate of $0.1$ and the use of the AdamW optimizer (Loshchilov & Hutter, 2019) with a weight decay of $10^{-6}$. To enhance robustness, random horizontal flips are applied as a data augmentation strategy.

**CelebA.** We adopted an unconditional DDPM framework similar to that used for CIFAR-10, but modified the architecture to handle 64×64 inputs. To better capture the higher complexity of CelebA, we scaled up the U-Net model to $118.8M$ parameters. All training settings, including the noise

variance schedule, optimizer configurations, and overall training procedure, were kept consistent with those used in the CIFAR-10 setup.

### D.3 LDS EVALUATION SETUP

For the LDS evaluation, we sample $M = 64$ random subsets $\mathbb{S}_m$ from the training set, with each subset comprising $50\%$ of the data (i.e. $\alpha = 0.5$). For each subset, three models are trained using different random seeds to improve robustness. We then compute the linear datamodeling score for each sample of interest as the Spearman rank correlation between the model's output and the attribution score as described in Eq. (8). In particular, to compute the simple loss $\mathcal{L}_{\text{simple}}(\boldsymbol{x}, \theta)$ as defined in Eq. (3) for any sample of interest, we evaluate it over 1000 timesteps uniformly spaced in the interval $[1, T]$. At each timestep, we further approximate the expectation $\mathbb{E}_{\epsilon}$ by sampling three standard Gaussian noise $\epsilon \sim \mathcal{N}(0, \mathbf{I})$. Finally, we average the LDS scores across the validation and generation samples to obtain the overall performance.

### D.4 BASELINES

We compare our proposed method against two major categories of the attribution methods: (1) nonparametric similarity-based method, which do not rely on model parameters and instead operate directly on the image features and representations, and (2) post-hoc retraining-free methods, which leverage gradient-based representations of a trained model to estimate the attribution score without requiring model retraining. Within the first category, we include two similarity-based methods: Raw Pixel and CLIP (Radford et al., 2021). Within the second category, we further divide the methods into *(gradient-based methods without kernels)* and *(gradient-based methods with kernels)*. For the gradient-based methods without kernels, we include the techniques such as Gradient (Charpiat et al., 2019), TracInCP (Pruthi et al., 2020) and GAS (Hammoudeh & Lowd, 2022). As for the gradient-based methods with kernel, we compare with some representative methods including TRAK (Park et al., 2023a) and D-TRAK (Zheng et al., 2024).

**Raw Pixel.** This is a naive similarity-based attribution method that directly uses the raw image as the feature representation. The attribution score is then computed by measuring the similarity—such as the dot product or cosine similarity—between the query sample and each training sample in the dataset.

**CLIP Similarity.** We adopt CLIP (Radford et al., 2021) as a baseline attribution method. Each target sample and training sample is embedded using CLIP, and the cosine similarity or the dot product of normalized embeddings between the two sample is taken as the attribution score. This provides a simple way to estimate the influence of training data, independent of the diffusion model or the LDS measurement function.

**Gradient.** This method is a gradient-based influence estimator from (Charpiat et al., 2019), where the attribution score for each training sample is computed using either the dot product or cosine similarity between its gradient(with respect to the model parameters) and that of a given test sample.

$$\tau(\boldsymbol{x}, \boldsymbol{z}^{(n)}; \mathbb{S}) = \mathcal{P}^{\top} \nabla_{\theta} \mathcal{L}_{\text{Simple}}(\boldsymbol{x}, \theta^*)^{\top} . \mathcal{P}^{\top} \nabla_{\theta} \mathcal{L}_{\text{Simple}}(\boldsymbol{z}^{(n)}, \theta^*)$$

$$\tau(\boldsymbol{x}, \boldsymbol{z}^{(n)}; \mathbb{S}) = \frac{\mathcal{P}^{\top} \nabla_{\theta} \mathcal{L}_{\text{Simple}}(\boldsymbol{x}, \theta^*)^{\top} . \mathcal{P}^{\top} \nabla_{\theta} \mathcal{L}_{\text{Simple}}(\boldsymbol{z}^{(n)}, \theta^*)}{\left\| \mathcal{P}^{\top} \nabla_{\theta} \mathcal{L}_{\text{Simple}}(\boldsymbol{x}, \theta^*) \right\| \left\| \mathcal{P}^{\top} \nabla_{\theta} \mathcal{L}_{\text{Simple}}(\boldsymbol{z}^{(n)}, \theta^*) \right\|} .$$

where $\mathcal{P}$ is the Gaussian random projection matrix that projects the gradient into low-dimensional subspace.

**TracInCP.** We adopt the TracInCP estimator introduced by (Pruthi et al., 2020) formulated as $\tau(\boldsymbol{x}, \boldsymbol{z}^{(n)}; \mathbb{S}) = \frac{1}{C} \Sigma_{c=1}^{C} \left( \mathcal{P}_c^{\top} \nabla_{\theta} \mathcal{L}_{\text{Simple}}(\boldsymbol{x}, \theta^c) \right)^{\top} \cdot \left( \mathcal{P}_c^{\top} \nabla_{\theta} \mathcal{L}_{\text{Simple}}(\boldsymbol{z}^{(n)}, \theta^c) \right)$ where $C$ denotes the number of model checkpoints uniformly sampled from the training trajectory, and $\theta^c$ represents the parameters at the $c$-th checkpoint. We use four checkpoints for each experiment. For example, on CIFAR-2, we select checkpoints at epochs $\{50, 100, 150, 200\}$.

**GAS.** This is a "renormalized" version of the TracInCP that leverages cosine similarity in place of raw dot products (Hammoudeh & Lowd, 2022).

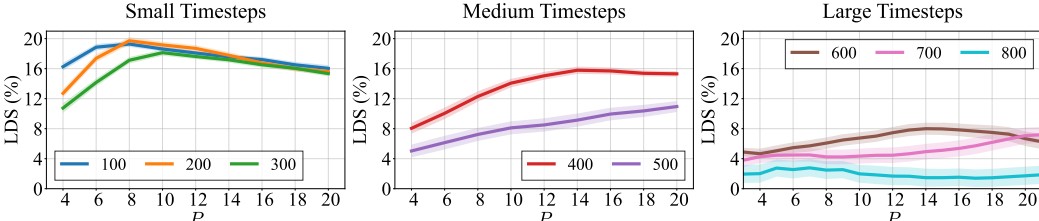

Figure 9: The LDS (%) on the validation set of CIFAR-2 across different patch sizes and timesteps on the low-resolution scale. The left panel shows LDS scores at early (small) timesteps ($t = 100$, 200, 300), where moderate patch sizes yield the highest scores. The middle and right panels show results at late (large) timesteps ($t = 600, 700, 800$), where smaller patches tend to underperform due to overwhelming noise, and larger patches offer relatively more stable performance.

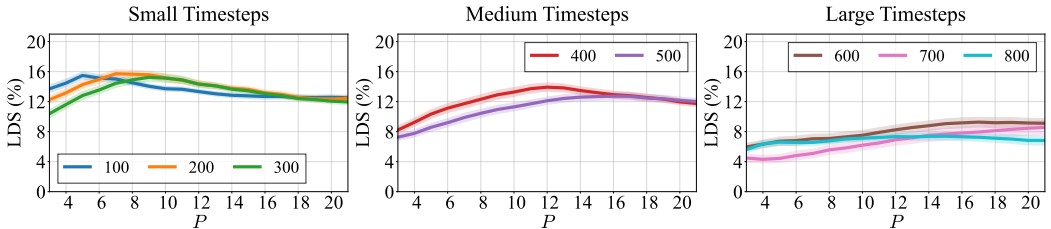

Figure 10: LDS (%) on the CelebA validation set across different patch sizes and timesteps at the original resolution. **Left**: small timesteps ($t \leq 300$), where moderate patch sizes achieve the highest scores. **Middle**: medium timesteps ($t = 400, 500$), where larger patches outperform smaller ones due to higher noise. **Right**: large timesteps, where noise dominates and informative signal is minimal.

**TRAK.** As discussed in (Zheng et al., 2024), TRAK (Park et al., 2023a) can be extended to the diffusion setting and the retraining-free TRAK can be implemented as:

$$\Phi_{\text{TRAK}} = \left[\phi\left(\boldsymbol{x}^1\right), \cdots, \phi\left(\boldsymbol{x}^N\right)\right]^{\top}, \text{where } \phi(\boldsymbol{x}) = \mathcal{P}^{\top}\nabla_\theta \mathcal{L}_{\text{Simple}}(\boldsymbol{x}, \theta^*)$$

$$\tau(\boldsymbol{x}, \boldsymbol{z}^{(n)}; \mathbb{S}) = \mathcal{P}^{\top}\nabla_\theta^* \mathcal{L}_{\text{Simple}}(\boldsymbol{x}, \theta^*)^{\top} \cdot \left(\Phi_{\text{TRAK}}^{\top}\Phi_{\text{TRAK}} + \lambda I\right)^{-1} \cdot \mathcal{P}^{\top}\nabla_\theta \mathcal{L}_{\text{Simple}}\left(\boldsymbol{z}^{(n)}, \theta^*\right)$$

where the $\lambda I$ is included for numerical stability and regularization effect.

**D-TRAK.** Similar to TRAK,

$$\Phi_{\text{D-TRAK}} = \left[\phi\left(\boldsymbol{x}^1\right), \cdots, \phi\left(\boldsymbol{x}^N\right)\right]^{\top}, \text{where } \phi(\boldsymbol{x}) = \mathcal{P}^{\top}\nabla_\theta \mathcal{L}_{\text{Square}}(\boldsymbol{x}, \theta^*)$$

$$\tau(\boldsymbol{x}, \boldsymbol{z}^{(n)}; \mathbb{S}) = \left(\mathcal{P}^{\top}\nabla_\theta \mathcal{L}_{\text{Simple}}(\boldsymbol{x}, \theta^*)\right)^{\top} \cdot \left(\Phi_{\text{TRAK}}^{\top}\Phi_{\text{TRAK}} + \lambda I\right)^{-1} \cdot \mathcal{P}^{\top}\nabla_\theta \mathcal{L}_{\text{Square}}\left(\boldsymbol{z}^{(n)}, \theta^*\right)$$

where $\lambda I$ is also included for numerical stability and regularization, as in TRAK and here $\mathcal{L}_{\text{Square}} = \|\epsilon_\theta(\boldsymbol{x}_t, t)\|^2$. Additionally, the output function $\mathcal{L}_{\text{Square}}$ could be replaced to other functions.

## E  ABLATION STUDIES

**Patch size selection.** For completeness, we also study the effect of patch size by evaluating LDS across $P \in [3, 21]$ at different timesteps on CelebA. To isolate timestep effects, we fix $\mathcal{T}$ to a single $t$ and vary $P$. Figure 10 shows LDS as a function of $P$ for each $t$. The trend mirrors CIFAR-2: the optimal patch size generally grows with $t$. At eraly timesteps ($t \leq 300$), small to moderate patches (e.g., $P = 5, 7, 9$) yield the highest scores, indicating local patterns dominate when noise is low. At mid-range timesteps ($400 \leq t \leq 500$), larger patches perform better (peaks shift to $P = 11, 13$), likely due to the need for more contextual information under higher noise. In the high-noise regime ($t \geq 600$), LDS drops and curves flatten across all patch sizes, suggesting informative signal is limited.

**Multiscale influence.** We further compare the optimal patch size between the original and the low-resolution settings on CIFAR-2. As shown in Figure 5 and Figure 9, at the early timestep $t=100$,

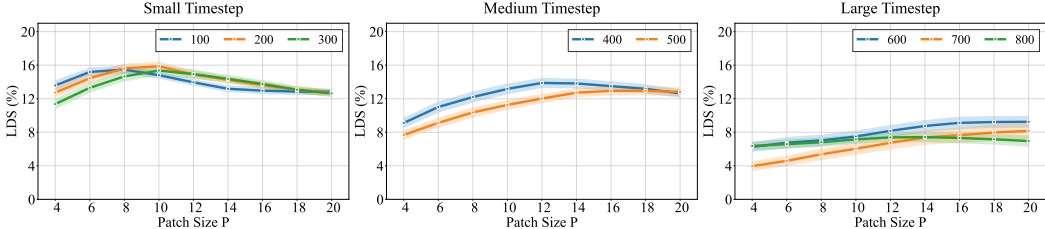

Figure 11: LDS (%) on the CelebA validation set across different patch sizes and timesteps at the low resolution. **Left**: small timesteps ($t{\leq}300$), where moderate patch sizes achieve the highest scores. **Middle**: medium timesteps ($t{=}400, 500$), where larger patches outperform smaller ones due to higher noise. **Right**: large timesteps, where noise dominates and informative signal is minimal.

the optima differ notably: the original resolution peaks at $P{=}5$, whereas the low-resolution curve peaks at $P{=}8$, indicating that coarser inputs benefit from slightly larger spatial context. By contrast, at $t = 200$ and $t = 300$, the two resolutions behave very similarly: on the original scale it peaks at $P = 7$ and $P = 9$ while low-resolution peaks at $P = 8$ and $P = 10$, with near-overlapping curves around their maxima. At mid timesteps ($t = 400, 500$), both resolution favors large patch size with $P = 21$. These results show that downscaling mainly shifts optimal to large patch size $P$ at small timestep, while mid/large timesteps show qualitatively similar trends across resolutions.

## F    VISUALIZATION

We provide additional visualizations, including counterfactual examples and proponent–opponent analyses.

### F.1    COUNTERFACTUAL VISULIZATION

We include more counterfactual visualizations in Figures 12 and 13. As shown, NDA identifies training examples whose removal produces a marked change in the output of a model retrained with the same seed. In Figure 12 (right column, third row), models retrained after removing images selected by Random and CLIP still synthesize an "automobile". In contrast, removing images selected by NDA or D-TRAK yields an image that resembles a mixture of "automobile" and "horse". Notably, the image after NDA's removal shows clearer horse morphology (e.g., head and leg outlines) than that after D-TRAK. These results suggest that NDA is an effective nonparametric approach for identifying training images with strong influence on a given target.

### F.2    PROPONENTS AND OPPONENTS VISULIZATION

Following Pruthi et al. (2020), we call training examples with positive influence scores proponents and those with negative scores opponents. For each target, we retrieve and visualize the top-5 proponents and the top-3 opponents. Qualitative results on CIFAR-2, CIFAR-10, and CelebA are shown in Figures 14, 15, and 16. Across datasets, NDA consistently retrieves proponents that are visually closer to the target than those selected by CLIP, and its opponents are correspondingly more dissimilar.

## G    ADDITIONAL ARCHITECTURES AND TRAINING OBJECTIVES

We further evaluate NDA under two new settings, varying both model architectures and training objectives:

1. **DiT-DDPM**: The target model uses a DiT architecture and is trained with the standard DDPM loss $\mathcal{L}_{\text{Simple}}$ in Eq. (3).

2. **DiT-Flow Matching**: The target model uses a DiT architecture and is trained with the rectified-flow loss $\mathcal{L}_{\text{FM}}$.

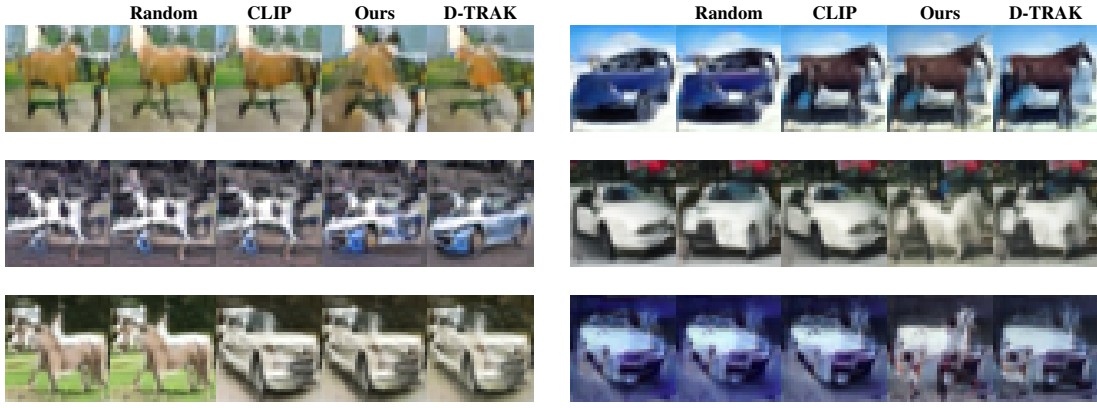

Figure 12: Counterfactual visualization on CIFAR-2 dataset. We compare samples generated by retrained models with different attribution methods using the same seed.

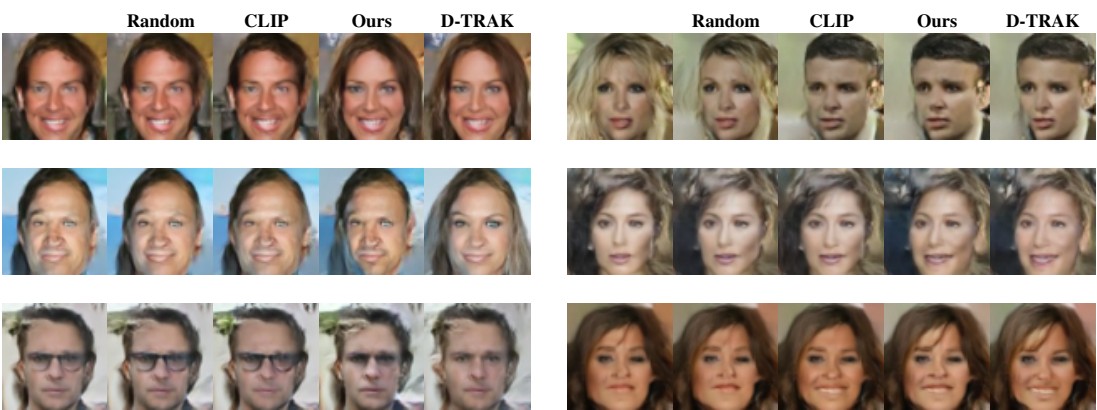

Figure 13: Counterfactual visualization on CelebA dataset. We compare samples generated by retrained models with different attribution methods using the same seed.

Table 2 reports LDS results for NDA and the baselines under the **DiT-DDPM** and **DiT-Flow Matching** settings described above. NDA consistently outperforms the Raw pixel and CLIP baselines and remains close to D-TRAK in both cases, indicating that NDA produces attributions that remain consistent across different models and training objectives.

## H ADDITIONAL PATCH-WISE BASELINES

We include additional results on two patch-wise baselines. For both baselines, we extract local patches from the query image $x$ and each training image $z$ using the same setup as in Sec. 3.1. Given these patches:

1. **Patch-wise Raw pixel**: We compute cosine similarity between every patch of $x$ and every patch of $z$. For each patch of $x$, we select its top-$k$ most similar patches in $z$ and sum their scores, then aggregate these per-patch sums over all patches of $x$ to obtain an image-level similarity score.

2. **Patch-wise CLIP**: We encode each patch with CLIP, and then compute cosine similarities between patch embeddings of $x$ and $z$, and apply the same procedure as above to get an image-level score.

We tune the patch size $P \in \{5, 7, 9, 11\}$ and the top-$k$ parameter separately for each patch-wise baseline on the CIFAR-2 validation dataset. For the patch-wise Raw pixel baseline, $P = 9$ and

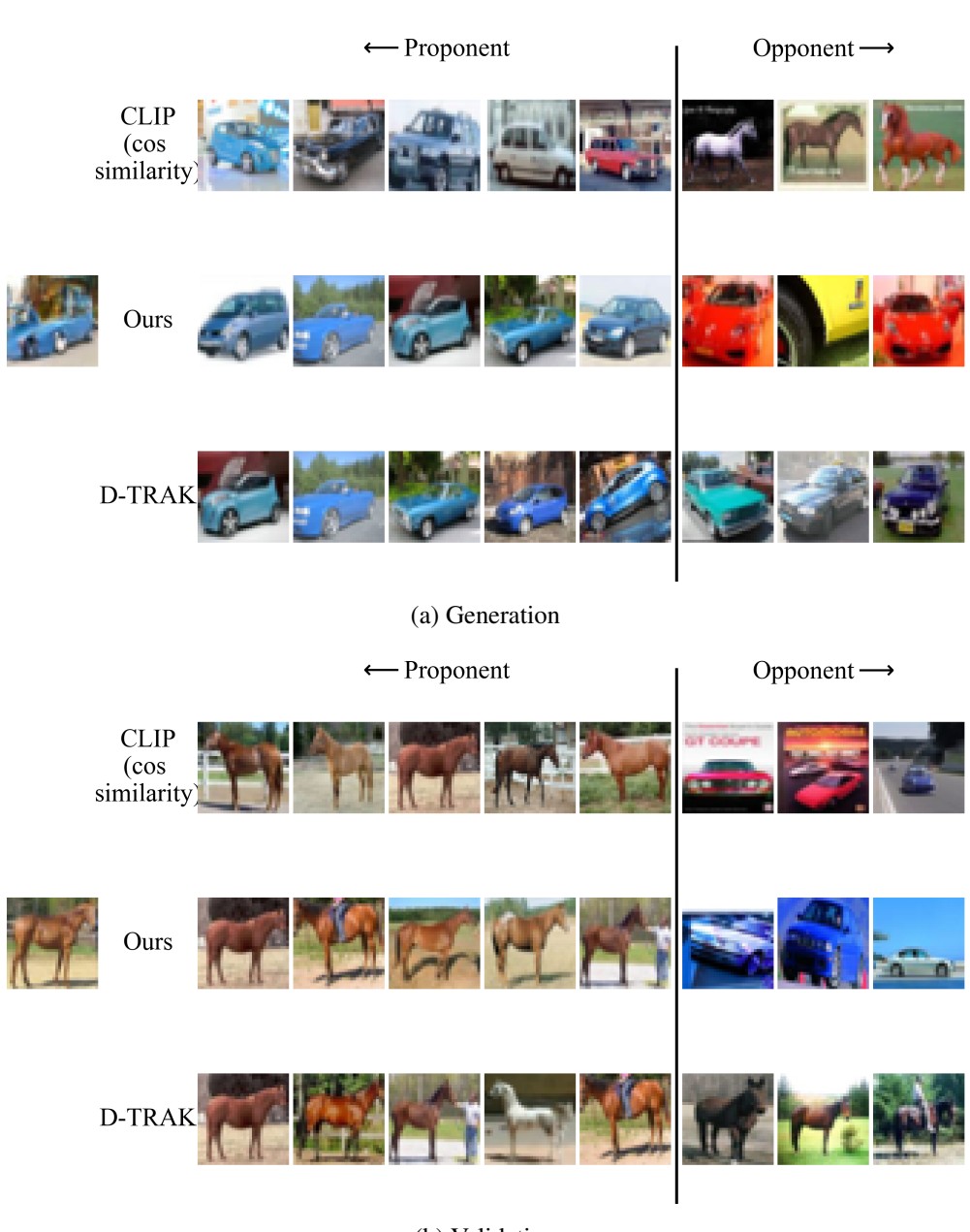

(a) Generation

(b) Validation

Figure 14: Proponents and opponents visualization on CIFAR-2 dataset using CLIP, NDA and D-TRAK using averaged timestep. For each sample of interest, we retrieve the most influential training samples based on attribution scores. 5 most positive influential training samples(left) and 3 most negative influential training samples(right) are shown.

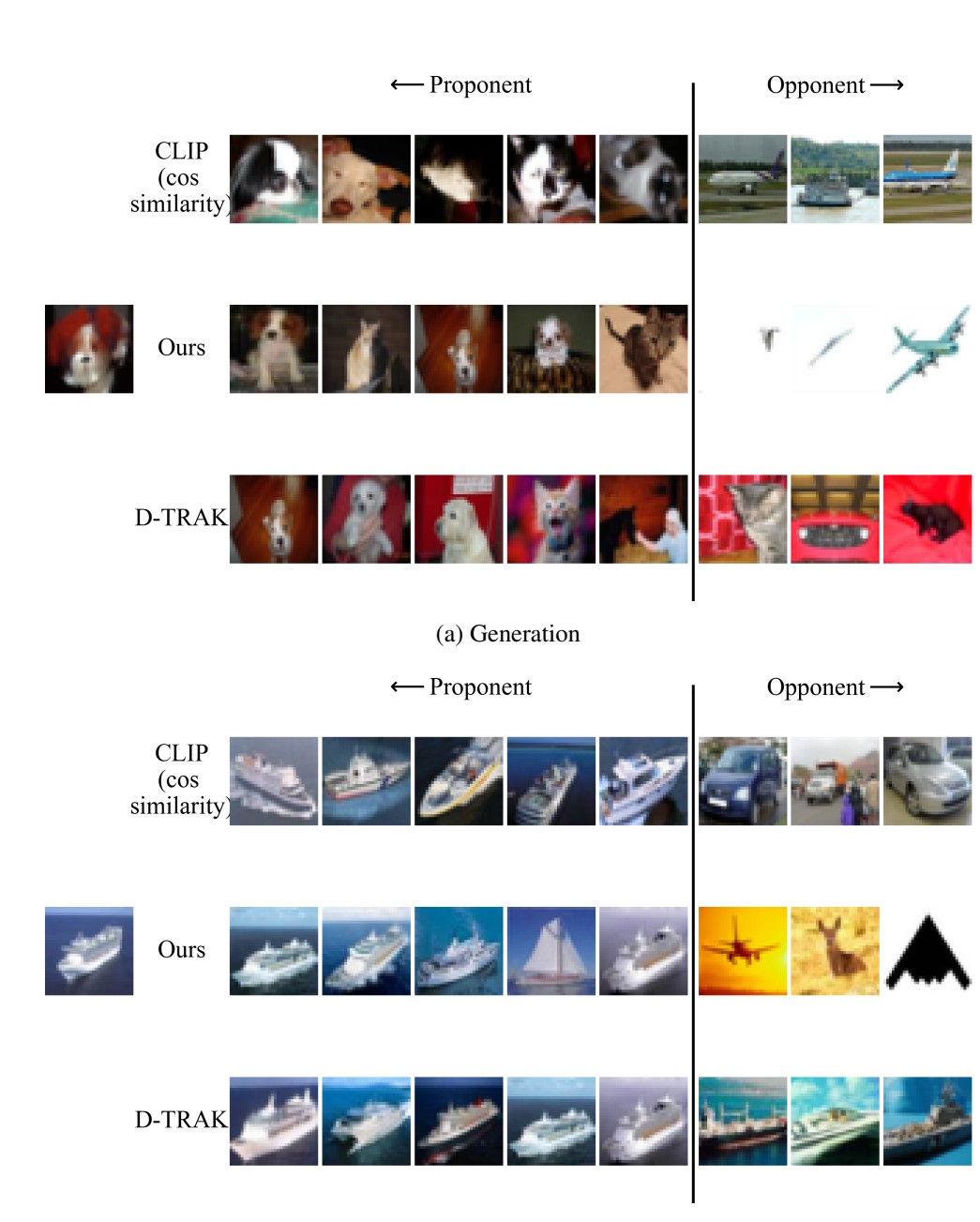

Figure 15: Proponents and opponents visualization on CIFAR-10 dataset using CLIP, NDA and D-TRAK using averaged timestep. For each sample of interest, we retrieve the most influential training samples based on attribution scores. 5 most positive influential training samples(left) and 3 most negative influential training samples(right) are shown.

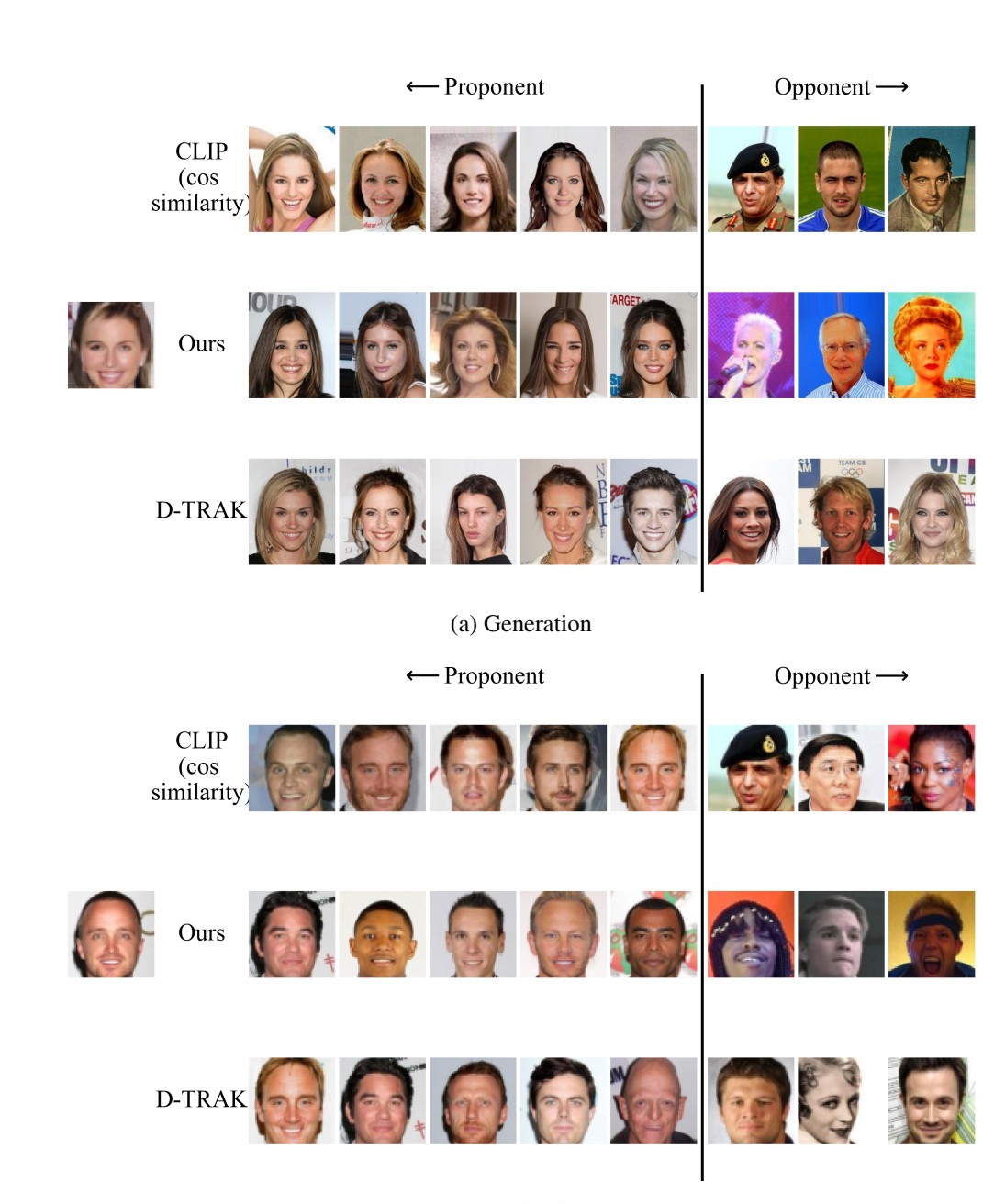

(a) Generation

(b) Validation

Figure 16: Proponents and opponents visualization on CelebA dataset using CLIP, NDA and D-TRAK using averaged timestep. For each sample of interest, we retrieve the most influential training samples based on attribution scores. 5 most positive influential training samples(left) and 3 most negative influential training samples(right) are shown.

| Setting    (Dataset) | Raw pixel (cos.) | CLIP (cos.) | NDA | D-TRAK |
|---|---|---|---|---|
| DiT–DDPM (CIFAR-2) | 10.81 | 4.32 | 17.13 | 21.17 |
| DiT–DDPM (CIFAR-10) | 1.73 | 5.13 | 6.07 | 6.62 |
| DiT–DDPM (CelebA) | 8.84 | 6.92 | 11.62 | 15.18 |
| DiT–Flow (CIFAR-2) | 3.95 | 5.30 | 15.61 | 15.20 |
| DiT–Flow (CIFAR-10) | 1.71 | 2.82 | 9.10 | 8.41 |

Table 2: LDS (%) on CIFAR-2, CIFAR-10, and CelebA validation datasets for DiT target models under DDPM and Flow Matching objectives.

| Method | CIFAR-2 | | CIFAR-10 | | CelebA | |
|---|---|---|---|---|---|---|
| | Validation | Generation | Validation | Generation | Validation | Generation |
| Patch-wise Raw pixel | 4.16 | 2.42 | 1.18 | 2.03 | 3.26 | 3.77 |
| Patch-wise CLIP | 6.61 | 4.73 | 1.20 | 1.61 | 6.35 | 5.76 |
| NDA | 24.88 | 15.91 | 11.81 | 7.41 | 16.89 | 13.92 |

Table 3: LDS (%) comparison between patch-wise Raw pixel / CLIP baselines and NDA on CIFAR-2, CIFAR-10, and CelebA.

$k = 100$ perform best; for the patch-wise CLIP baseline, $P = 5$ and $k = 100$ work best. We fix these choices for all datasets and report the corresponding results in Table 3.

As shown in Table 3, NDA consistently outperforms both patch-wise Raw pixel and patch-wise CLIP baselines across all datasets, suggesting that simply patchifying the images does not improve LDS and that NDA's gains arise from insights theoretically motivated by the analytical score, rather than from patchification alone.

