# OpenReview forum: "Nonparametric Data Attribution for Diffusion Models"
_ICLR.cc/2026/Conference — Submitted to ICLR 2026_

### Official Review · Reviewer_E9AP · 2025-10-21

**Soundness:** 2
**Presentation:** 2
**Contribution:** 2
**Rating:** 2
**Confidence:** 3

**Summary:**

The authors propose Nonparametric Diffusion Attribution (NDA), a gradient-free data-attribution method for diffusion models that scores patch-level influence using the analytic optimal score, and is optimized with convolution-based operations. They target black-box/proprietary settings with no access to a model's parameters, and show attributions that outperform prior baselines and approach gradient-based methods on CIFAR-10 and CelebA.

**Strengths:**

The paper seems practical and easy enough to follow in situations where no model access is available. The authors also report spatially interpretable results, and the method is shown to outperform significantly the tested prior baselines.

**Weaknesses:**

1. A few core assumptions and scope limits should be made explicit. Most derivations rely on naive score-matching or score identities for standard diffusion losses, so general statements about “model-agnostic” influence seem like an overreach.  For example, if we inject an exact-reconstruction penalty in the loss, the results may no longer hold.

2. Compute requirements seem concerning. The convolutional trick proposed by the authors helps with peak memory blow-up, but the per-image convolution over all training patches and multiple timesteps/scales won't scale very well.

3. The authors test their work in relatively low-resolution settings (up to 64×64).

4. The visual distribution tested a relatively simple, with generally smooth contours and simple patches.

**Questions:**

(number references to the Weekness section)

On (2). It would be valuable to report wall-clock vs. dataset size, have some theoretical scaling analysis on compute for data size/resolution, and give scaling guidances.

On (3). Are the multiscale tricks and hyperparameters stable at higher resolutions or for more scales?—an experiment at ≥256×256 would be very valuable.

On (4). How does the algorithm behave with more complex distributions (e.g. Imagenet)?

---

> ### Author Response · Authors · 2025-11-22
> **Response to Reviewer E9AP [1/2]**
>
> Thank you for your valuable review and suggestions. Below we respond to the comments in ***Weaknesses (W)*** and ***Questions (Q)***.
>
> ---
>
> ***W1: “Model-agnostic” assumptions should be clarified.***
>
> Thank you for raising this concern. We would like to clarify that the motivation from analytical scores of standard diffusion processes provides an intrinsic way to quantify data influence. As a result, its applicability is **not restricted to models trained with the standard diffusion loss**. To substantiate our claim, we provide additional results under two new settings, varying both model architectures and training objectives:
>
> 1. **DiT–DDPM**: The target model uses a DiT architecture and is trained with the standard DDPM loss $\\mathcal{L}\_{\\text{Simple}}$ in Eq. (3).
> 2. **DiT–Flow Matching**: The target model uses a DiT architecture and is trained with the rectified-flow loss $\\mathcal{L}\_{\\text{FM}}$ [1].
>
> For both settings, we retrain the models $\\theta\^{*}(\\mathbb{S}\_m)$ needed for LDS evaluation, following the procedure in Section 2.3. In the LDS evaluation for **DiT–Flow Matching**, we set $\\mathcal{F}=\\mathcal{L}\_{\\text{FM}}$. Note that in both cases, NDA is applied in exactly the same **model-free** manner as before, without accessing model parameters, gradients, or training objectives. By contrast, the gradient-based baseline D-TRAK uses per-example gradients of the corresponding training objectives evaluated on the target models.
>
> The results on the CIFAR-2 validation set are as follows (due to time limitation, results on other datasets will be included in the revision):
>
> | DiT-DDPM| Raw pixel (cosine) | CLIP (cosine) | NDA| D-TRAK|
> |---|:---:|:---:|:---:|:---:|
> | CIFAR2 |  10.81  | 4.32    |  17.13    |  21.17  |
>
>
> | DiT-Flow Matching| Raw pixel (cosine) | CLIP (cosine) | NDA| D-TRAK|
> |---|:---:|:---:|:---:|:---:|
> | CIFAR2 |   3.95    |  5.30     |  15.61    |  15.20  |
>
> These results show that NDA consistently outperforms raw pixel and CLIP, while approaching the performance of D-TRAK across both architectures and training objectives. This supports our claim that NDA produces attributions that remain consistent across different models and training objectives.
>
> ---
>
> ***W2 & Q2: Compute cost and lack of scaling analysis.***
>
> We appreciate this feedback. NDA is primarily a preliminary attempt at **nonparametric data attribution**, targeting scenarios where the model is not accessible (e.g., proprietary) and providing a model-independent notion of influence (see discussion in Section 3.5). Therefore, direct comparisons of computational cost to model-dependent methods should be interpreted with this context in mind.
>
> Nevertheless, we provide wall-clock runtimes for attribution computation on a single NVIDIA A100 GPU using our PyTorch implementation:
>
> |         | #Samples | D-TRAK Train (hh:mm) | D-TRAK Attr. (hh:mm) | NDA (hh:mm) |
> | ------- | -------- | -------------------- | -------------------- | ----------- |
> | CIFAR2  | 5,000      | 00:29               | 00:44                 | 0:50        |
> | CIFAR10 | 50,000   | 03:35             | 06:20                 | 8:58       |
>
> For D-TRAK, both the training time of the target model and the attribution time (gradient + projection) are explicitly listed. More runtime results will be included in the revision.
>
> Algorithmically, for a batch of $B$ test images, $\|\\mathcal{T}\|$ timesteps, and $N$ training images of resolution $H \\times W$, the time complexity of our NDA implementation is $\\mathcal{O}(B \|\\mathcal{T}\| N (HW)\^2 P\^2)$, i.e., it scales linearly in $B$, $\|\\mathcal{T}\|$, and $N$, and quadratically in image size and patch size. Note that the computation of NDA is highly parallelizable (no deep hierarchical structures & all patches can be computed simultaneously), providing significant potential for acceleration.

---

> ### Author Response · Authors · 2025-11-22
> **Response to Reviewer E9AP [2/2]**
>
> ***W3 & Q3: Limited to low resolution and unclear stability of multiscale tricks.***
>
> We provide additional evaluation on higher resolutions and clarify the behavior of our multiscale design. Following Zheng et al. [2], we extend our evaluation to the ArtBench-2 dataset, which contains $256 \times 256$ images. We keep the evaluation protocol unchanged and report LDS on validation sets below:
>
> | Raw Pixel (cosine) | CLIP (cosine) | Gradient (cosine) | TracInCP | GAS | **NDA (ours)** | TRAK|
> |--------------------|---------------|-------------------|---------|-----|------|----------------|
> | 2.58               | 8.62          | 7.22              | 9.69    | 9.65| **11.28**      | 12.26|
>
>
> As shown, NDA outperforms the raw-pixel and CLIP baselines on ArtBench-2, indicating that our method remains effective on high-resolution data.
> To evaluate the stability of our multiscale design, we further ablate NDA at $256\\times256$ over multiple timesteps on the validation set, comparing the original-resolution setting with its low-resolution and multiscale counterparts. We reuse the same multiscale hyperparameters  as in the CelebA $64\\times64$ experiments and accordingly scale the patch size by a factor of $4$, without additional tuning.
>
> | timestep     | 100   | 200   | 300   | 400   |
> |-----------------|-------|-------|-------|-------|
> | original resolution   |  9.35  |  8.18  | 7.40   | 5.93   |
> | low-resolution |  10.42  | 9.09   |  7.59  |  6.21  |
> | multiscale |  11.10  | 9.53   | 7.92   | 6.55   |
>
> The results show that multiscale attribution remains effective and hyperparameters are stable at higher resolutions.
>
> ---
>
> ***W4 & Q4: How does the algorithm behave with more complex distributions(e.g. ImageNet)?***
>
> Thank you for raising this point. To the best of our knowledge, existing data-attribution benchmarks (including gradient-based methods such as D-TRAK) generally do not evaluate on full ImageNet. We therefore follow the community practice and evaluate on the ArtBench dataset at 256x256 resolution, representing a standard benchmark for high resolution cases.
>
> Our new ArtBench results in ***W3 & Q3*** shows that NDA consistently outperforms strong nonparametric baselines (raw-pixel and CLIP), suggesting robust behavior on complex, high-resolution distributions.
>
> We are aware that CustomMark [3] reports experiments on ImageNet. However, these focus on proactive watermarking of selected classes rather than general per-image attribution, and thus are not directly comparable to our setting.
>
> ---
>
> ***References:***\
> [1] Flow matching for generative modeling. Yaron Lipman, et al. ICLR 2023. \
> [2] Intriguing properties of data attribution in diffusion models. Zheng, Xiaosen, et al. ICLR 2024.\
> [3] CustomMark: Customization of diffusion models for proactive attribution.Vishal Asnani, et al. ICCV 2025 workshop.

---

> > ### Comment · Reviewer_E9AP · 2025-11-25
> >
> > Thanks for addressing my feedback. For W1, it would be convincing to see results on other datasets.
> >
> > For the remaining questions, my concerns have mostly been addressed. I have raised my score accordingly.

---

> > > ### Author Response · Authors · 2025-11-25
> > > **Thank you for your feedback and raising the score**
> > >
> > > Thank you for your timely feedback and for raising the score. We have been actively running experiments on additional datasets to further validate ***W1***, and we will update the results as soon as they are ready, before the rebuttal period ends.

---

> > > ### Author Response · Authors · 2025-12-01
> > > **Additional results addressing W1**
> > >
> > > Following your suggestion, we have expanded our evaluation and now include results on **CIFAR-10** and **CelebA** using DiT backbones, in addition to the CIFAR-2 results above.
> > >
> > > For CIFAR-10, we use the same DiT architecture as in CIFAR-2 (32.47M parameters) and train it under two objectives: **DiT-DDPM** and **DiT-Flow Matching**. For CelebA, due to computational constraints, we focus on the **DiT-DDPM** setting and adopt a larger backbone: an unconditional DiT-DDPM model adapted to 64×64 resolution and scaled up to 95.46M parameters by increasing depth, hidden dimension, and attention heads.
> > >
> > > The results on these datasets are as follows:
> > >
> > > | DiT-DDPM | Raw pixel (cosine) | CLIP (cosine) | NDA | D-TRAK |
> > > |---|:---:|:---:|:---:|:---:|
> > > | CIFAR-2  | 10.81 | 4.32 | 17.13 | 21.17 |
> > > | CIFAR-10 |  1.73 | 5.13 |  6.07 |  6.62 |
> > > | CelebA   |  8.84 | 6.92 | 11.62 | 15.18 |
> > >
> > > | DiT-Flow Matching | Raw pixel (cosine) | CLIP (cosine) | NDA | D-TRAK |
> > > |---|:---:|:---:|:---:|:---:|
> > > | CIFAR-2  | 3.95 | 5.30 | 15.61 | 15.20 |
> > > | CIFAR-10 | 1.71 | 2.82 |  9.10 |  8.41 |
> > >
> > > Across all datasets, model sizes, and training objectives, NDA consistently outperforms raw-pixel and CLIP baselines and remains comparable to D-TRAK, **showing strong generality**.

---

### Official Review · Reviewer_m648 · 2025-10-29

**Soundness:** 3
**Presentation:** 3
**Contribution:** 2
**Rating:** 6
**Confidence:** 2

**Summary:**

The paper introduces a nonparametric approach to data attribution for diffusion models motivated by scalability. It computes patch-level influence scores derived from the optimal score-function formulation. Effectiveness is evaluated via Linear Datamodeling Score (LDS) and counterfactual removal-and-retrain experiments.

**Strengths:**

The paper is well motivated and well presented. The proposed approach is solid, although more explanations will be helpful (see weakness). In the evaluated scenarios, the methods seem to perform well.

**Weaknesses:**

1. While the method is motivated by a score-function view, the estimator still just behaves like a complicated version of a similarity measure against training patches. Additional ablations/theory to isolate what truly helps will be very appreciated. For example, do naive patch-level similarities across timesteps already recover most of the effect?
2. The paper is motivated by scalability. However, it seems that the paper still only evaluates on smaller models that generate low-res images.

**Questions:**

As the authors also point out, patch-level attribution seems to be interpretable. Do you think that for diffusion model kind image generation models, patch-level attribution is just fundamentally a better approach compared to image-level attribution, as it provides more fine-grained signals?

---

> ### Author Response · Authors · 2025-11-22
> **Response to Reviewer m648 [1/2]**
>
> Thank you for your valuable review and suggestions. Below we respond to the comments in ***Weaknesses (W)*** and ***Questions (Q)***. In the revision, we also include additional evaluations under two new settings (DiT-DDPM and DiT-Flow Matching), which further show that our method **remains effective and consistent across different model architectures and training objectives**.
>
> ---
>
> ***W1: Do naive patch-level similarities across timesteps recover most of the effect?***
>
> Thank you for raising this concern. To test this, we implement two naive patch-wise influence baselines (c.f. Eq. (11)) that directly measure patch-level distance across timesteps, without mirroring the analytical score function:
> 1. **Patch L2 similarity**: $\\tau(\\boldsymbol{x}\_{t,\\Omega\_{\ell}}, \\boldsymbol{u}; \\mathbb{P}\_{\\Omega}(\\mathbb{S})) = -\\|\\boldsymbol{x}\_{t, \\Omega\_{\ell}} - \\boldsymbol{u} \\|^2$
> 2. **Noise-weighted patch L2 similarity**: $\\tau(\\boldsymbol{x}\_{t,\\Omega\_{\ell}}, \\boldsymbol{u}; \\mathbb{P}\_{\\Omega}(\\mathbb{S})) = -\\frac{\\|\\boldsymbol{x}\_{t, \\Omega\_{\ell}} - \\sqrt{\\bar{\\alpha}\_{t}} \\boldsymbol{u} \\|^2}{2(1 - \\bar{\\alpha}\_t)}$
>
> We keep the same aggregation procedure from patch-level influence scores to image-level attribution (i.e., Eq. (12)), and report results averaged over timesteps $\\mathcal{T} = \\{100, 200, 300, 400, 500\\}$:
>
> | Method                         | CIFAR-2 Val | CIFAR-2 Gen | CIFAR-10 Val | CIFAR-10 Gen | CelebA Val | CelebA Gen |
> |--------------------------------|-----------:|-----------:|------------:|------------:|-----------:|-----------:|
> | Patch L2 similarity   |     5.94   |     3.91   |      1.60   |      1.67   |     7.81   |     7.07   |
> | Noise-weighted patch L2 similarity    |     5.34   |     3.15   |      1.53   |      1.82   |     7.43   |     6.58   |
> | **NDA (ours)**   | **24.88**  | **15.91**  |   **11.81** |    **7.41** |  **16.89** | **13.92** |
>
> As a complementary check (suggested by other reviewers), we also design two patch-wise baselines based on other similarities. Specifically, we extract local patches from the query image $\\boldsymbol{x}$ and each training image $\\boldsymbol{z}$ using the setup as in Sec. 3.1. Given these patches:
>
> 1. **Patch-wise Raw pixel**: We compute **cosine similarity** between every patch of $\\boldsymbol{x}$ and every patch of $\\boldsymbol{z}$. For each patch of $\\boldsymbol{x}$, we select its top-$k$ most similar patches in $\\boldsymbol{z}$ and **sum** their scores, then **aggregate** these per-patch sums over all patches of $\\boldsymbol{x}$ to obtain an image-level similarity score.
>
> 2. **Patch-wise CLIP**:  We encode each patch with CLIP, compute **cosine similarities between patch embeddings** of $\\boldsymbol{x}$ and $\\boldsymbol{z}$, and apply the same procedure as above to get an image-level score.
>
> We tune the patch size $P \\in \\{5, 7, 9, 11\\}$ and the top-$k$ parameter separately for each patch-wise baseline on the CIFAR-2 validation dataset. For the patch-wise Raw pixel baseline, $P = 9$ and $k = 100$ perform best; for the patch-wise CLIP baseline, $P = 5$ and $k = 100$ work best. We fix these choices for all datasets and report the corresponding results in the table below:
>
> | Method                | CIFAR-2 Val | CIFAR-2 Gen | CIFAR-10 Val | CIFAR-10 Gen | CelebA Val | CelebA Gen |
> | :-------------------- | :--------: | :--------: | :---------: | :---------: | :--------: | :--------: |
> | Patch-wise Raw pixel       |   4.16      | 2.42      |1.18         | 2.03        | 3.26         |   3.77     |
> | Patch-wise CLIP |    6.61         | 4.73     | 1.20       | 1.61         | 6.35       | 5.76         |
> | NDA  |  24.88         | 15.91      |11.81          | 7.41      | 16.89      | 13.92        |
>
> Taken together with the ablations above, these results show that **NDA’s gains arise from its insights theoretically motivated by the analytical score, rather than from patch extraction alone**.
>
> ---
>
> ***W2: Lack of experiments on larger / higher-resolution models.***
>
> Thank you for pointing this out. Following Zheng et al. [1], we extend our evaluation to the ArtBench-2 dataset, which contains $256\times256$ images. We keep the evaluation protocol unchanged and report LDS on the validation dataset:
>
> | Raw pixel (cosine) | CLIP (cosine) | Gradient (cosine) | TracInCP | GAS | **NDA (ours)** | TRAK|
> |--------------------|---------------|-------------------|---------|-----|------|----------------|
> | 2.58               | 8.62          | 7.22              | 9.69    | 9.65| **11.28**      | 12.26 |
>
>
> As shown in the table, NDA outperforms the Raw pixel and CLIP baselines on ArtBench-2 and remains competitive with gradient-based methods (Gradient, TracInCP, GAS, TRAK), indicating that our method remains effective on high-resolution datasets.

---

> ### Author Response · Authors · 2025-11-22
> **Response to Reviewer m648 [2/2]**
>
> ***Q1: Is patch-level attribution fundamentally better than image-level attribution for diffusion models?***
>
> Thanks for your insightful question. We agree that patch-level attribution is a natural way to obtain more fine-grained signals. However, our experiments (see response to ***W1***) show that patchification alone is not sufficient, suggesting that NDA’s gains come mostly from its insights theoretically motivated by the analytical score.
>
> ---
>
> ***References:***\
> [1] Intriguing properties of data attribution in diffusion models. Zheng, Xiaosen, et al. ICLR 2024.

---

### Official Review · Reviewer_iVsD · 2025-11-03

**Soundness:** 3
**Presentation:** 3
**Contribution:** 3
**Rating:** 4
**Confidence:** 4

**Summary:**

The paper proposes a method for quantifying the relevancy of individual training samples to the outputs generated by diffusion models without requiring model access or gradients. The method performs patch-level similarity comparisons between generated and training images to quantify the attribution. Evaluation results using CIFAR-2, CIFAR-10, and CelebA datasets, show that NDA performance is close to gradient-based approaches and outperforming existing nonparametric baselines.

**Strengths:**

1. Proposing an efficient query-only/nonparametric, data attribution approach for diffusion models.
2. The method is based on analytical properties of the diffusion score function.
3. The patch-level mapping can be used for understanding the relationship between training and generated images.
4. Experiments show consistent improvements over the baselines.

**Weaknesses:**

1. Figure 1 – The proposed method is not clear from the figure. I suggest improving the figure.
2. No clear description of what are the requirements for performing the attribution
3. Not clear how the ground-truth was generated?
4. Experiments should be extended to larger and more complex datasets (e.g., ImageNet) to prove scalability and generalization.
5. Comparisons with other strong nonparametric or model-free baselines is missing, for example:

[a] CustomMark: Customization of Diffusion Models for Proactive Attribution
[b] Montrage: Monitoring training for attribution of generative diffusion models

6. Method's runtime analysis is missing.

**Questions:**

1. Please explain what are the requirements for performing the attribution?
2. How the ground-truth was generated?

---

> ### Author Response · Authors · 2025-11-22
> **Response to Reviewer iVsD [1/2]**
>
> Thank you for your valuable review and suggestions. Below we respond to the comments in ***Weaknesses (W)*** and ***Questions (Q)***. We also include, in our ***General Response***, additional evaluations under two new settings (DiT-DDPM and DiT-Flow Matching), which further show that our method **remains effective and consistent across different model architectures and training objectives**.
>
> ---
>
> ***W1: Figure 1 is not clear.***
>
> Thank you for the suggestion. We have updated Figure 1 to provide a clearer and more explicit illustration of the proposed NDA method.
>
> ---
>
> ***W2 & Q1: The requirements for performing the distribution.***
>
> Our NDA method is *model-free*: it does ***not*** require access to the target generative model, its parameters, gradients, architecture, or training procedure.
>
> To perform attribution, NDA **only requires access to the training dataset**. Given a query image $\\boldsymbol{x}$, all computations are performed directly between $\\boldsymbol{x}$ and the training images $\\boldsymbol{z}\^{(n)}$, based solely on patch-level influence derived from the analytical score function.
>
> It is worth noting that NDA does not require knowledge of the target model’s noise schedule. In our experiments, we simply adopt the standard DDPM forward process with 1000 steps and a linear beta schedule, regardless of what noise schedule or objective was used to train the target model.
>
> ---
>
> ***W3 & Q2: How was the ground-truth generated?***
>
> In our paper, the “ground-truth” influence refers to **the counterfactual ground truth defined by LDS**. To evaluate data attribution methods, we follow the LDS framework of Park et al. [1], which provides a principled way to measure the true influence of training examples via counterfactual retraining. Below, we summarize how LDS defines ground truth:
>
> Concretely, for each dataset we sample $M = 64$ random training subsets $\\mathbb{S}\_m \\subset \\mathbb{S}$, each containing 50% of the training data, and train a separate diffusion model $\\theta\^{\*}({\\mathbb{S}\_m})$ on each subset. For a test input $\\boldsymbol{x}$, we then compute the set of model outputs
> $$
> \\{\\mathcal{F}(\\boldsymbol{x};\\theta\^*(\\mathbb{S}\_m)) : m \\in [M]\\},
> $$
> where $\\mathcal{F}$ denotes the model output function used in LDS (for diffusion models we set $\\mathcal{F} = \mathcal{L}\_{\\text{simple}}$, i.e., the DDPM training loss). Intuitively, if a subset $\\mathbb{S}\_m$ contains examples that are more influential for $\\boldsymbol{x}$, a model trained on $\\mathbb{S}\_m$ should explain $\\boldsymbol{x}$ better and thus yield a lower loss.
>
> The ordering of the subsets $\\{\\mathbb{S}\_m\\}$ induced by these losses is treated—by definition of LDS—as the **ground-truth ranking** of subset influence. Our attribution method produces its own predicted ranking via the aggregated scores $g\_{\\tau}(\\boldsymbol{x}, \\mathbb{S}\_m; \\mathbb{S})$; LDS then measures the Spearman rank correlation between these two rankings.
>
> ---
>
> ***W4: Experiments should be extended to larger and more complex datasets.***
>
> Thank you for the suggestion. To assess NDA on the higher-resolution dataset, we follow Zheng et al. [2] and extend our evaluation to the ArtBench-2 dataset, which contains $256\\times256$ images. We keep the evaluation protocol unchanged and report LDS on the validation dataset:
>
>
> | Raw pixel (cosine) | CLIP (cosine) | Gradient (cosine) | TracInCP | GAS | **NDA (ours)** | TRAK|
> |--------------------|---------------|-------------------|---------|-----|------|----------------|
> | 2.58               | 8.62          | 7.22              | 9.69    | 9.65| **11.28**      | 12.26|
>
>
> As shown in the table, NDA outperforms the Raw pixel and CLIP baselines on ArtBench-2 and remains competitive with gradient-based methods (Gradient, TracInCP, GAS, TRAK), indicating that our method remains effective on high-resolution datasets.

---

> ### Author Response · Authors · 2025-11-22
> **Response to Reviewer iVsD [2/2]**
>
> ***W5: Comparisons with other strong nonparametric or model-free baselines.***
>
> Thank you for highlighting the importance of strong nonparametric baselines. Our work focuses on a **model-agnostic** setting, where the attribution method cannot modify the training pipeline and does not access model parameters or gradients. By contrast, CustomMark [3] fine-tunes pretrained text-to-image diffusion models to embed concept-specific (e.g., artist- or class-specific) watermarks into generated images, which are later decoded for attribution; MONTRAGE [4] modifies the diffusion training pipeline to log internal representations and trains additional attribution heads on top of them. Both methods require **modifying and retraining the generator or the training pipeline**, and therefore fall outside our model-agnostic setting.
>
> Within the model-agnostic regime, CLIP similarity is the most widely used baseline. As suggested by other reviewers, we add two new **patch-wise** baselines: (i) Patch-wise Raw pixel and (ii) Patch-wise CLIP. For both baselines, we extract local patches from the query image $\\boldsymbol{x}$ and each training image $\\boldsymbol{z}$ using the same setup as in Sec. 3.1. Given these patches:
>
> 1. **Patch-wise Raw pixel**: We compute cosine similarity between every patch of $\\boldsymbol{x}$ and every patch of $\\boldsymbol{z}$. For each patch of $\\boldsymbol{x}$, we select its top-$k$ most similar patches in $\\boldsymbol{z}$ and **sum** their scores, then **aggregate** these per-patch sums over all patches of $\\boldsymbol{x}$ to obtain an image-level similarity score.
>
> 2. **Patch-wise CLIP**: We encode each patch with CLIP, and then compute cosine similarities between patch embeddings of $\\boldsymbol{x}$ and $\\boldsymbol{z}$, and apply the same procedure as above to get an image-level score.
>
> We tune the patch size $P \\in \\{5, 7, 9, 11\\}$ and the top-$k$ parameter separately for each patch-wise baseline on the CIFAR-2 validation dataset. For the patch-wise Raw pixel baseline, $P = 9$ and $k = 100$ perform best; for the patch-wise CLIP baseline, $P = 5$ and $k = 100$ work best. We fix these choices for all datasets and report the corresponding results in the table below:
>
> | Method                | CIFAR-2 Val | CIFAR-2 Gen | CIFAR-10 Val | CIFAR-10 Gen | CelebA Val | CelebA Gen |
> | :-------------------- | :--------: | :--------: | :---------: | :---------: | :--------: | :--------: |
> | Patch-wise Raw pixel       |   4.16      | 2.42      |1.18         | 2.03        | 3.26         |   3.77     |
> | Patch-wise CLIP |    6.61         | 4.73     | 1.20       | 1.61         | 6.35       | 5.76         |
> | NDA  |  24.88         | 15.91      |11.81          | 7.41      | 16.89      | 13.92        |
>
> The results suggest that NDA is a strong nonparametric attribution method and the superior performance arises from the insights theoretically motivated by the analytical score, rather than from simple patchification.
>
> ---
>
> ***W6: Runtime analysis for the method.***
>
> We provide wall-clock runtimes for attribution computation on a single NVIDIA A100 GPU using our PyTorch implementation:
>
> |         | #Samples | D-TRAK Train (hh:mm) | D-TRAK Attr. (hh:mm) | NDA (hh:mm) |
> | ------- | -------- | -------------------- | -------------------- | ----------- |
> | CIFAR2  | 5,000      | 00:29               | 00:44                 | 0:50        |
> | CIFAR10 | 50,000   | 03:35             | 06:20                 | 8:58       |
>
> For D-TRAK, both the training time of the target model and the attribution time (gradient + projection) are explicitly listed. More runtime results will be included in the revision.
>
> We would also like to note that NDA is primarily a preliminary attempt at **nonparametric data attribution**, targeting scenarios where the model is not accessible (e.g., proprietary) and providing a model-independent notion of influence (see discussion in Section 3.5). Therefore, direct comparisons of computational cost to model-dependent methods should be interpreted with this context in mind.
>
> ---
>
> ***References:***\
> [1] TRAK: Attributing model behavior at scale. Sung Min Park, et al. PMLR 2023. \
> [2] Intriguing properties of data attribution in diffusion models. Zheng, Xiaosen, et al. ICLR 2024.\
> [3] CustomMark: Customization of diffusion models for proactive attribution. Vishal Asnani, et al. ICCV workshop 2025 \
> [4] MONTRAGE: Monitoring training for attribution of generative diffusion models. Jonathan Brokman, et. al. ECCV 2024.

---

### Official Review · Reviewer_Hezj · 2025-11-04

**Soundness:** 2
**Presentation:** 3
**Contribution:** 2
**Rating:** 4
**Confidence:** 4

**Summary:**

This paper introduces Nonparametric Diffusion Attribution (NDA), a novel, model-agnostic method for data attribution in diffusion models. The key insight is to reinterpret the weighting term from the analytical, optimal score function of a diffusion process as an influence score. By grounding this in a local score function, the method derives a patch-wise similarity metric that is computationally efficient and requires no access to model parameters. The method demonstrates strong empirical performance on the Linear Datamodeling Score (LDS) when compared to several baseline methods.

**Strengths:**

1. Novel Non-Parametric Framework: The paper makes a valuable contribution by framing the attribution problem from a non-parametric, data-driven perspective.
2. Theoretical Motivation: The method is motivated by reinterpreting the weighting term (W_t) from the analytical form of the optimal score function (Eq. 5-6) as a measure of influence. This provides a strong theoretical justification for why this specific form of similarity should be effective.
3. Strong Empirical Performance (vs. Chosen Baselines): On the LDS and counterfactual retraining metrics, the proposed NDA method is shown to be highly effective. It substantially outperforms other full-image non-parametric baselines (Raw Pixel, CLIP).

**Weaknesses:**

1. Unsupported Claim of Model-Independence: The paper makes a strong, highly valuable claim that "our empirical results show that the attribution scores remain consistent across a variety of architectures and training regimes" (Page 6, Line 281). However, there are no empirical results presented in the manuscript (including the appendix) to substantiate this.

2. Missing "Apples-to-Apples" Baselines: The paper's core method (NDA) is patch-based. However, the non-parametric baselines it is compared against ("Raw Pixel" and "CLIP Similarity") are full-image-based. This is an "apples-to-oranges" comparison that inflates the perceived contribution of NDA. The strong performance seen might not come from the specific "score-function" formulation but simply from the act of using patches, which is a known technique. Some "naive patch-based" baselines include:
    * Patch-wise Raw Pixel: A method that computes L2 similarity on top-k raw patches and aggregates them.
    * Patch-wise Feature Similarity: A method that uses a standard feature extractor (e.g., CLIP, a pretrained CNN, or DINOv2) to embed patches and then computes a top-k aggregated similarity.

3. Failure to Disentangle Causal Factors: The paper's novelty rests on two assumptions: theoretical motivation via optimal score function and localization. The current experiments do not disentangle these. It is unclear whether the method's good performance comes from the "locality" assumption (Eq. 9) or the "optimal score" formula itself (Eq. 6). The authors should provide an ablation study for an "Image-wise NDA" baseline. This method would be derived directly from the full-image optimal score function (Eq. 5) and would use the full-image weighting term $W_t$ (Eq. 6) as the influence score, with no patch decomposition.

**Questions:**

See weaknesses.

---

> ### Author Response · Authors · 2025-11-22
> **Response to Reviewer Hezj [1/2]**
>
> Thank you for your valuable review and suggestions. Below we respond to the comments in ***Weaknesses (W)***.
>
> ---
>
> ***W1: Unsupported claim that NDA’s attribution scores are model-independent across architectures and training regimes.***
>
> Thank you for raising this concern. To substantiate our claim, we provide additional results under two new settings, varying both model architectures and training objectives:
>
> 1. **DiT–DDPM**: The target model uses a DiT architecture and is trained with the standard DDPM loss $\\mathcal{L}\_{\\text{Simple}}$ in Eq. (3).
> 2. **DiT–Flow Matching**: The target model uses a DiT architecture and is trained with the rectified-flow loss $\\mathcal{L}\_{\\text{FM}}$ [1].
>
> For both settings, we retrain the models $\\theta\^{*}(\\mathbb{S}\_m)$ needed for LDS evaluation, following the procedure in Section 2.3. In the LDS evaluation for **DiT–Flow Matching**, we set $\\mathcal{F}=\\mathcal{L}\_{\\text{FM}}$. Note that in both cases, NDA is applied in exactly the same **model-free** manner as before, without accessing model parameters, gradients, or training objectives. By contrast, the gradient-based baseline D-TRAK uses per-example gradients of the corresponding training objective evaluated on the target model.
>
> The results on the CIFAR-2 validation set are as follows (due to time limitation, results on other datasets will be included in the revision):
>
> | DiT-DDPM| Raw pixel (cosine) | CLIP (cosine) | NDA| D-TRAK|
> |---|:---:|:---:|:---:|:---:|
> | CIFAR-2 |  10.81  | 4.32    |  17.13    |  21.17  |
>
>
> | DiT-Flow Matching| Raw pixel (cosine) | CLIP (cosine) | NDA| D-TRAK|
> |---|:---:|:---:|:---:|:---:|
> | CIFAR-2 |   3.95    |  5.30     |  15.61    |  15.20  |
>
> These results show that NDA consistently outperforms Raw pixel and CLIP, while approaching the performance of D-TRAK across both architectures and training objectives. This supports our claim that **NDA produces attributions that remain consistent across different models and training objectives**.
>
> ---
>
> ***W2: Missing apples-to-apples patch-based baselines (Raw pixel and CLIP).***
>
> Thank you for the suggestion. We add additional results on two patch-wise baselines. For both baselines, we extract local patches from the query image $\\boldsymbol{x}$ and each training image $\\boldsymbol{z}$ using the same setup as in Sec. 3.1. Given these patches:
>
> 1. **Patch-wise Raw pixel**: We compute cosine similarity between every patch of $\\boldsymbol{x}$ and every patch of $\\boldsymbol{z}$. For each patch of $\\boldsymbol{x}$, we select its top-$k$ most similar patches in $\\boldsymbol{z}$ and **sum** their scores, then **aggregate** these per-patch sums over all patches of $\\boldsymbol{x}$ to obtain an image-level similarity score.
>
> 2. **Patch-wise CLIP**: We encode each patch with CLIP, and then compute cosine similarities between patch embeddings of $\\boldsymbol{x}$ and $\\boldsymbol{z}$, and apply the same procedure as above to get an image-level score.
>
> We tune the patch size $P \\in \\{5, 7, 9, 11\\}$ and the top-$k$ parameter separately for each patch-wise baseline on the CIFAR-2 validation dataset. For the patch-wise Raw pixel baseline, $P = 9$ and $k = 100$ perform best; for the patch-wise CLIP baseline, $P = 5$ and $k = 100$ work best. We fix these choices for all datasets and report the corresponding results in the table below:
>
> | Method                | CIFAR-2 Val | CIFAR-2 Gen | CIFAR-10 Val | CIFAR-10 Gen | CelebA Val | CelebA Gen |
> | :-------------------- | :--------: | :--------: | :---------: | :---------: | :--------: | :--------: |
> | Patch-wise Raw pixel       |   4.16      | 2.42      |1.18         | 2.03        | 3.26         |   3.77     |
> | Patch-wise CLIP |    6.61         | 4.73     | 1.20       | 1.61         | 6.35       | 5.76         |
> | NDA  |  24.88         | 15.91      |11.81          | 7.41      | 16.89      | 13.92        |
>
> As shown in the table, NDA consistently outperforms both patch-wise Raw pixel and patch-wise CLIP baselines across all datasets. These results suggest that simply patchifying the images do not improve LDS and that NDA’s gains arise from **its insights theoretically motivated by the analytical score, rather than from patchification alone**.

---

> > ### Author Response · Authors · 2025-11-22
> > **Response to Reviewer Hezj [2/2]**
> >
> > ***W3: Failure to disentangle causal factors / lack of an “Image-wise NDA” baseline.***
> >
> > Thank you for the suggestion. We have implemented an **Image-wise NDA** baseline that directly applies the optimal score formulation in Eq. (5) at the full-image level: we compute the weights $W\_t(\\boldsymbol{z}\^{(n)} \\mid \\boldsymbol{x}\_t)$ from Eq. (6) and use them as image-level influence scores, **without** any patch decomposition. All other settings (noise schedule, chosen timesteps $\\mathcal{T}$, and evaluation protocol) are kept identical to our patch-wise NDA, and we report the performance in the table below:
> >
> > | Method   | CIFAR-2 Val        | CIFAR-2 Gen  | CIFAR-10 Val   | CIFAR-10 Gen       | CelebA Val        | CelebA Gen        |
> > | :---------------------- | :---------------: | :---------------: | :---------------: | :---------------: | :---------------: | :---------------: |
> > | Image-wise NDA (Eq. 6)  | 9.96      | 5.01      | 6.07         | 2.27        | 5.82          |  3.93        |
> > | NDA (ours)   | 24.88      | 15.91      | 11.81        | 7.41           | 16.89     | 13.92   |
> >
> > As shown in the table, the image-wise NDA baseline is clearly weaker than our NDA method in the paper. This is expected, as the ideal score in Eq. (5) does not induce any generalization beyond the training set, while NDA is inspired by the analytical score under locality and equivariance assumptions, which provide the necessary inductive bias for stronger attribution.
> >
> > ---
> >
> > ***References***\
> > [1] Flow matching for generative modeling. Yaron Lipman, et al. ICLR 2023.

---

> ### Author Response · Authors · 2025-12-01
> **Additional results addressing W1**
>
> As a follow-up to our previous W1 response, we now report additional results on CIFAR-10 and CelebA using DiT backbones.  These experiments further validate our claim that NDA’s attribution scores remain consistent across different architectures and training regimes.
>
> For CIFAR-10, we use the same DiT architecture as in CIFAR-2 (32.47M parameters) and train it under two objectives: **DiT-DDPM** and **DiT-Flow Matching**. For CelebA, due to computational constraints, we focus on the **DiT-DDPM** setting and adopt a larger backbone: an unconditional DiT-DDPM model adapted to 64×64 resolution and scaled up to 95.46M parameters by increasing depth, hidden dimension, and attention heads.
>
> The results on these datasets are as follows:
>
> | DiT-DDPM | Raw pixel (cosine) | CLIP (cosine) | NDA | D-TRAK |
> |---|:---:|:---:|:---:|:---:|
> | CIFAR-2  | 10.81 | 4.32 | 17.13 | 21.17 |
> | CIFAR-10 |  1.73 | 5.13 |  6.07 |  6.62 |
> | CelebA   |  8.84 | 6.92 | 11.62 | 15.18 |
>
> | DiT-Flow Matching | Raw pixel (cosine) | CLIP (cosine) | NDA | D-TRAK |
> |---|:---:|:---:|:---:|:---:|
> | CIFAR-2  | 3.95 | 5.30 | 15.61 | 15.20 |
> | CIFAR-10 | 1.71 | 2.82 |  9.10 |  8.41 |
>
> Across all datasets, model sizes, and training objectives, NDA consistently outperforms raw-pixel and CLIP baselines and remains comparable to D-TRAK, **showing strong generality**.

---

### Author Response · Authors · 2025-11-22
**Summary of Paper Revision**

We thank all reviewers for their constructive feedback, and we have responded to each reviewer individually. We have also uploaded a **Paper Revision** including additional results:


- $\\textrm{\\color{blue} Figure 1.}$: Updated to provide a clearer and more explicit illustration of the proposed NDA method.


- $\textrm{\color{blue}{Appendix H}}$: Additional LDS results for  DiT target model trained with two different objectives (DDPM loss and Flow Matching loss).


- $\textrm{\color{blue}{Appendix I}}$: Additional LDS results for patch-wise raw-pixel and CLIP baselines.

---

### Author Response · Authors · 2025-12-02
**Summary of Rebuttal Discussion**

We thank all reviewers for their constructive feedback. During the rebuttal, we conducted additional experiments and clarified several aspects of NDA to address the main concerns raised by each reviewer.

As the discussion phase ended unexpectedly early, **Reviewers Hezj, iVsD**, and **m648** were unable to follow up further on our rebuttal.

**Reviewer E9AP** reacted positively to our rebuttal and **raised the score**, while also requesting more evidence for W1. We subsequently added additional experiments on CIFAR-10 and CelebA, which are now included in our follow-up response.

Overall, we believe the discussion was constructive and that the main concerns have been effectively addressed through new results and clarifications.

---

### Meta-Review · Area_Chair_8rkZ · 2025-12-16

**Summary:**

The paper proposes a nonparametric approach (DNA) to data attribution for diffusion models without access to the model and gradients.

Strengths identified by reviewers include, its novel Framework, strong performance, and well-motivation.

The reviewers raise several concerns and the major concern is the lack of clear comparisons to verify the contribution of the core proposed components such as patch-wise nonparametric baselines and lack of experiments to substantiate that the attribution scores remain consistent across a variety of architectures and training regimes. Another concern is that the lack of evaluation on a larger and more complex dataset such as ImageNet and also larger model for evaluating its scalability and generalization. Though the authors have provided a detailed feedback with additional results with new baselines and a dataset with a larger resolution, these concerns are not fully addressed.

Given these considerations, this paper fails to meet the standards of ICLR. The authors are encouraged to address the highlighted issues to strengthen their contribution to the field of Data attribution for Diffusion Models.

**Reviewer Concerns:**

The authors provided detailed feedback and most of concerns have been addressed including, 1) Missing "Apples-to-Apples" Baselines; 2) Failure to Disentangle Causal Factors; 3) unclear figure, and description of requirements and groundtruth generation; 4) runtime analysis; 5) limited to low-resolution images.

However, several concerns including, a) lack of evaluation on larger and more complex datasets for evaluating scalability and generalization; b) Though additional results with DiT with DDPM and rectified-flow are added, the core claim of the paper is that attribution scores remain consistent across a variety of architectures and training regimes  which is not fully substantiated.

**Reviewer Scores:**

The initial reviewer scores are mixed (two borderline reject, one reject and one borderline accept). Reviewer E9AP has confirmed raising the scores while other reviewers would have maintained the scores

---

### Decision · Program_Chairs · 2026-01-26

Reject